# DiPmark: A Stealthy, Efficient and Resilient Watermark for Large Language Models

## Abstract

Watermarking techniques offer a promising way to secure data via embedding covert information into the data. A paramount challenge in the domain lies in preserving the distribution of original data during watermarking. Our research extends and refines existing watermarking framework, placing emphasis on the importance of a distribution-preserving (DiP) watermark. Contrary to the current strategies, our proposed DiPmark preserves the original token distribution during watermarking (stealthy), is detectable without access to the language model API or weights (efficient), and is robust to moderate changes of tokens (resilient). DiP-mark operates by initially selecting a random assortment of 'green' tokens prior to the generation of a word, then employing a distribution-preserving reweighting function to enhance the utilization of these green tokens. The empirical benchmarks of our approach underscore its stealthiness, efficiency, and resilience, making it a robust solution for watermarking tasks that demand impeccable quality preservation.

## 1 Introduction

In an age where artificial intelligence has transcended the barriers of human-like text generation, the validity and origin of such content have come under significant scrutiny. The ascent of large language models (LLMs) and their uncanny capability to emulate human prose present a double-edged sword. On one hand, the potential applications are vast, from simplifying complex tasks to augmenting human capabilities. Conversely, there exists a latent threat of misuse, as observed in scenarios ranging from students seeking undue advantages in academic settings to the propagation of misinformation on digital platforms such as Reddit, Twitter, and StackOverflow.

As the demarcation between machine-generated and human-penned content blurs, the conventional heuristic methods to differentiate between them falter. The recent failures of even state-of-the-art detectors epitomize the dire need for a more robust solution. In this context, *watermarking* emerges as a nuanced solution: the deliberate embedding of distinctive signals within generated content to assert its provenance. By marking content at its source, this technique provides a mechanism to trace the origin of any disseminated information. Beyond the widely held notion that watermarks should be identifiable via a secret key, there exist several fundamental characteristics necessary for an efficient watermark within language models:

- (Stealthy) The watermark should integrate seamlessly into the generated content, preserving the distribution and quality of the original text, making it impossible to discern between watermarked and non-watermarked text.

- (Efficient) With the secret key, detecting watermark within the content (or a fragment of it) should be fast and straightforward without accessing the language models and prompts.

- (Resilient) If the content undergoes moderate changes or modifications, the watermark should remain intact and identifiable.

To the best of our understanding, current watermarking techniques do not adhere to the aforementioned three key properties. These methods either impact the model's sampling distribution, leading to reduction in the model's performance (Kirchenbauer et al., 2023), lack resilience against text alterations such as editing or cropping (Christ et al., 2023), necessitate thousands of inference step during detection process (Kuditipudi et al., 2023), or require the token logits of language model API

**Table 1:** Characteristics of the watermark proposed in this work, compared with previous works.

| Characteristics | Kirchenbauer et al. (2023) | Christ et al. (2023) | Kuditipudi et al. (2023) | Hu et al. (2023) | DiPmark |
|---|---|---|---|---|---|
| Stealthy (Sec. 4 & 6.1) | ✗ | ✓ | ✓ | ✓ | ✓ |
| Efficient (Sec. 5 & 6.2) | ✓ | ✓ | ✗ | ✗ | ✓ |
| Resilient (Sec. 5 & 6.3) | ✓ | ✗ | ✓ | ✓ | ✓ |

during detection (Hu et al., 2023), resulting in huge computational costs. Conventional steganographic methods, which insert subtle embedding within language model-generated texts, often fall short in terms of detectable and resilient (Ziegler et al., 2019; Kaptchuk et al., 2021). Our endeavor is the creation of pioneering watermarks for text attribution to a language model that encompass all three crucial properties (see Table 1 for an overall comparison).

Our watermarking framework, in alignment with pre-existing schema Kirchenbauer et al. (2023), is essentially comprised of two components: a generate function, which transforms a prompt and a proprietary watermark key into a sample derived from the language model, and a detect function that aligns a potential watermarked text with the secret key. During the text generation process, language model providers will reweight the output probability of the current token using a confidential key. The reweight function maintains stealth, ensuring that each instance of text generation aligns with the original language model's distribution. In the detection phase, the presence of a watermark can be efficiently detected by solely using the secret key and the watermarked text, potentially supplemented with standard hash functions found in Python. We also validate the above mentioned three fundamental properties of our watermark via experimental assessments on major language models, including BART-large model (Liu et al., 2020) and LLaMA-2 (Touvron et al., 2023).

**Significance of preserving text distribution in watermarking.** We emphasize the critical significance of preserving the distribution of text in watermarking, driven by the following compelling justifications: a) Stealthy Watermarking: A watermark that disrupts the original distribution of a language model lacks the attribute of stealthiness. Such alterations make it relatively straightforward to distinguish between watermarked and unwatermarked LMs through multiple instances of sampling. b) Industry-Level LLM Application: When contemplating the application of a watermark to industry-standard LLMs like ChatGPT and Bard, the primary consideration is to ensure that the watermark does not compromise the performance of these foundational LLMs. Any watermark that interferes with the original text distribution will inevitably impact the quality of generated text, an outcome that is deemed unacceptable by industry stakeholders.

**Our contributions.** Our work tackles the problem of designing watermarks for large language models without affecting its natural performance and advances the state-of-the-art in multiple ways.

- We propose a novel watermarking framework, DiPmark, that introduces a **provably** distribution-preserving watermarking scheme for language models. Unlike existing methods, DiPmark is **simultaneously** stealthy, efficient, and resilient.

- We develop an effective watermark detection mechanism for DiPmark. By utilizing hypothesis testing and a well-defined statistical approach, we can reliably detect the presence of the watermark while maintaining a low false positive rate. We also show our detect algorithm is **provably** robust against text modifications.

- Experimentally, we validate the distribution-preserving property of DiPmark by evaluating the generated text quality on machine translation and text summarization tasks. Notably, the detection time for 1,000 watermarked sequences produced by LLaMA-2 stands at a mere 90 seconds without the need of API access. Furthermore, DiPmark exhibits robustness even when subjected to 20% to 30% random text modifications, encompassing insertions, deletions, and substitutions.

## 2   RELATED WORK

In a recent seminal work, Kirchenbauer et al. (2023) introduced a pioneering watermarking scheme tailored for LLMs. However, this approach inevitably leads to a pivotal change in the distribution of the generated text, potentially compromising the quality of the generated content. To maintain an unaltered output distribution in watermarked content, alternative strategies have been explored. Christ et al. (2023) and Kuditipudi et al. (2023) employed the inverse sampling method to generate watermarked token distributions. Notably, Christ et al. (2023)'s method faces resilience issues

under modifications or changes and lacks empirical validation for detectability. Meanwhile, Kuditipudi et al. (2023)'s approach necessitates the secret key distribution during detection, potentially compromising data security and watermark stealthiness. Moreover, their detection process involves thousands of resampling steps from the secret key distribution, which is inefficient for lengthy texts. Hu et al. (2023) used inverse sampling and permutation based reweight methods for watermarking, but the detector requires the token logits of language model API, undermining its operational efficiency. A detailed discussion of watermarking LLMs is in Appendix C.

Our research aligns closely with Kirchenbauer et al. (2023). In their methodology, they employed watermarking for text derived from a language model by separating the token set into 'red' and 'green' lists. Building on this foundation, our research retains the red-green list configuration, but introduces an evolved family of reweighting strategies. This approach ensures: 1) a promoted utilization of green tokens, and 2) equivalency in distribution between a sample from the watermarked language model and one from the original language model.

## 3 PRELIMINARY

**Notations.** To lay a foundation for our ensuing discussion, we first establish a few essential notations. Let's represent the vocabulary (or token) set by $V$ and its size or volume by $N = |V|$. We further introduce the set $\mathcal{V}$, defined as an aggregation of all conceivable string sequences, even accounting for those of zero length. In the context of a language model, it produces a token sequence based on a predetermined context. For a single step of this process, the likelihood of generating the imminent token $x_{n+1} \in V$, while considering the current context—spanning from $x_1$ to $x_n$—is represented as $P_M(x_{n+1} \mid x_1, x_2, ..., x_n)$. Inherent to its design, the LLM adopts an autoregressive mode of operation. This implies that the combined probability of producing several tokens, specifically from $x_{n+1}$ to $x_{n+m}$ takes the form:

$$P_M(x_{n+1}, \ldots, x_{n+m} \mid x_1, x_2, ..., x_n) = \prod_{i=1}^{m} P_M(x_{n+i} \mid x_1, x_2, ..., x_n, x_{n+1}, \ldots, x_{n+i-1}).$$

For the sake of brevity and clarity, we opt for the condensed notation: $P_M(\boldsymbol{x}_{n+1:n+m} \mid \boldsymbol{x}_{1:n})$, where $\boldsymbol{x}_{n+1:n+m} = (x_{n+1}, \ldots, x_{n+m})$. Note that the prompt is deliberately omitted in this representation.

In the context of watermarking, the server provider will use a set of *i.i.d. watermark cipher* $\{\theta_i \in \Theta, i \in \mathbb{N}\}$ on the cipher space $\Theta$ to generate the text. The cipher $\theta_i$ is usually generated by a secret key $k$ and a fragment of the previous context, named *texture key*, $\boldsymbol{s}_i$. Each $\theta_i$ is independent and following the same distribution $P_\Theta$. The watermarked output of the LLM follows distribution $P_W(x_{n+1} \mid \boldsymbol{x}_{1:n}, \theta_n)$, which is conditioned on both the cipher $\theta_n$ and the context $\boldsymbol{x}_{1:n}$. We name the $P_W$ as the *reweight strategy* of the watermark.

**Red-Green list watermark.** Building upon the soft/hard red-green list delineated in Kirchenbauer et al. (2023), the vocabulary set is systematically bifurcated into the red and green lists. This process subtly accentuates the usage of green tokens during sampling. Specifically, for generating the $n$-th token $x_n$, the red and green lists are fashioned by randomly permuting the vocabulary set $V$. Then, a predefined separator is subsequently interposed into the permuted list, such that the first $\gamma$ percent tokens of the permuted list constitute the red list, whereas the remaining $1 - \gamma$ percent form the green list. In this way, the cipher space $\Theta$ encompasses all potential permutations of $V$, with $\theta_n$ signifying a distinct permutation of $V$. During token generation, the likelihood of green list tokens is augmented to watermark the resultant text. For detection purposes, a user, equipped with the secret key, can deduce the red-green list and ascertain watermark presence through the discrepancy in the red and green token counts.

**Reweight strategies.** The reweight strategy stands as the linchpin of the watermark algorithm, fundamentally shaping the distribution of watermarked text. As introduced in Kirchenbauer et al. (2023), the authors propose a soft reweight technique. Given an initial token probability $p(t)$, the watermarked probability for the token, denoted by $p'(t)$, is formulated as:

$$p'(t) = \begin{cases} \dfrac{p(t)}{\sum_{t \in \text{red}} p(t) + \sum_{t \in \text{green}} e^\delta p(t)}, & t \in \text{red list}; \\ \dfrac{e^\delta p(t)}{\sum_{t \in \text{red}} p(t) + \sum_{t \in \text{green}} e^\delta p(t)}, & t \in \text{green list}, \end{cases} \tag{1}$$

where $\delta > 0$ is a predetermined constant. A cursory examination of this strategy reveals an inherent bias in the watermarked distribution. For example, consider $\gamma = 0.5$, suggesting that half of $V$ comprises the red list. With $V = \{a, b\}$, and given probabilities $p(a) = 0.99$ and $p(b) = 0.01$, there are two equivalent permutations of $V$ with congruent appearance likelihoods. An analysis for any value of $\delta > 0$ yields $p'(a) = 0.5(\frac{e^{\delta}p(a)}{e^{\delta}p(a)+p(b)} + \frac{p(a)}{e^{\delta}p(b)+p(a)}) < p(a)$. This underscores that the red-green list watermark does not retain the original text's probability.

**Distribution preserving.** A reweight strategy, denoted $P_W$, is said to be distribution-preserving at an individual generation step if, for all $\boldsymbol{x}_{1:n} \in \mathcal{V}$ and any $i \leq n$, it holds that $P_M(x_i|\boldsymbol{x}_{1:i-1}) = \mathbb{E}_{\theta_i \sim P_{\Theta}}[P_W(x_i|\boldsymbol{x}_{1:i-1}, \theta_i)]$. Further, if a watermark framework retains the inherent text distribution throughout all generation steps, i.e., for all sequences $\boldsymbol{x}_{1:n} \in \mathcal{V}$ and any positive integer $n$, we have $P_M(\boldsymbol{x}_{1:n}) = \mathbb{E}_{\theta_1, ..., \theta_n}[P_W(\boldsymbol{x}_{1:n}|k, \theta_1, ..., \theta_n)]$, then the watermark is deemed distribution-preserving. It is pivotal to recognize that a distribution-preserving reweight strategy can naturally lead to a distribution-preserving watermark, as illustrated by:

$$\mathbb{E}_{\theta_1, ..., \theta_n}[P_W(\boldsymbol{x}_{1:n}|k, \theta_1, ..., \theta_n)] = \mathbb{E}_{\theta_1, ..., \theta_n}\left[\prod_{i=1}^{n} P_W(x_i|\boldsymbol{x}_{1:i-1}, k, \theta_i)\right]$$
$$= \prod_{i=1}^{n} \mathbb{E}_{\theta_i}[P_W(x_i|\boldsymbol{x}_{1:i-1}, k, \theta_i)] = P_M(\boldsymbol{x}_{1:n}). \tag{2}$$

The above equality stems from the independence property of the set $\{\theta_i\}$. Therefore, to establish a distribution-preserving watermark, it is essential to incorporate both: a) a distribution-preserving reweight strategy and b) an *i.i.d.* set of ciphers, $\{\theta_i\}$.

In our research, we introduce a watermarking technique with a "distribution-preserving" characteristic. This attribute guarantees that the text distribution remains unaltered even as we enhance the utilization of tokens from the green list during the watermarking process.

## 4 DiPmark

**Motivation.** The reweight strategy presented in Kirchenbauer et al. (2023) disrupts the inherent text distribution when advocating for the green tokens during sampling. This disruption can lead to skewed sampling, having a pronounced impact on the quality of the resultant text. Our objective is to conceptualize a novel reweight strategy that ensures the token distribution remains unaltered during the watermarking process. Contrary to the approach in Kirchenbauer et al. (2023) that promotes the use of all tokens from the green list, we emphasize increasing the *sum of the probability* of the green-list tokens. In this way, the watermarked text, when exposed to the secret key, will still exhibit a bias towards the green-list tokens. By adopting this idea, we design a special reweight function, which preserves the text distribution during watermarking.

**Cipher space for watermarking.** Our chosen watermark cipher space encompasses the permutations of the vocabulary set, denoted as $\Theta = \{V_1^p, ..., V_{N!}^p\}$, wherein $V_i^p$ represents a permutation of $V$. For the cipher distribution $P_{\Theta}$, we employ a uniform distribution over $\Theta$, signifying that each permutation is equally probable for selection.

$\alpha$**-reweight.** Let $\theta \in \Theta$ be a cipher, constituting a permutation of $V$. The probabilities of individual tokens can be organized within the interval $[0, 1]$ according to their respective positions in $\theta$. Given a fixed constant $\alpha$ in $[0, 1]$, the token probabilities within the interval $[0, \alpha]$ are adjusted to 0, while those in the interval $[\alpha, 1]$ are scaled by a factor of $\frac{1}{1-\alpha}$. Through this procedure, we elevate the sum of the probability of green-list tokens for *arbitrary* permutation separator $\gamma$, as the green-list tokens consistently appear towards the end of the ordered set $\theta$. If we represent $\theta$ as $\{t_1, ..., t_N\}$ and denote $P_M(\cdot|\boldsymbol{x})$ as the original token distribution, let $F(i|\theta) = \frac{1}{1-\alpha} \max\{\sum_{j=1}^{i} P_M(t_j|\boldsymbol{x}) - \alpha, 0\}$. The resultant $\alpha$-reweight probability distribution is expressed as $P_W(t_i|\boldsymbol{x}, \theta) = F(i|\theta) - F(i-1|\theta)$.

We wish to highlight the distinction between the probability quantile $\alpha$ and the red-green list separator $\gamma$. In accordance with the definition in Kirchenbauer et al. (2023), $\gamma$ serves as the partition for the permuted token list. In contrast, $\alpha$ separates the *probability interval* $[0, 1]$ of the permuted token list. Thus, both the $\alpha$-reweight and DiP-reweight (as subsequently defined) remain irrelevant to $\gamma$, while still effectively promoting the utilization of green list tokens.

---

**Algorithm 1** DiPmark generator

---

1: **Input:** key for watermark $k \in K$, reweight parameter $\alpha$, prompt $\boldsymbol{x}_{-m:0}$, generate length $n \in \mathbb{N}$, texture key history $hist$, and permutation generation function $h$.
2: **for** $i = 1, \ldots, n$ **do**
3:     Calculate the LM distribution for generating the $i$-th token $P_M(\cdot \mid \boldsymbol{x}_{-m:i-1})$.
4:     Generate a texture key $\boldsymbol{s}_i$ based on $\boldsymbol{x}_{1:i-1}$.
5:     **if** $\boldsymbol{s}_i \in hist$ **then**
6:         Sample the next token $x_i$ using original LM distribution $P_M(\cdot \mid \boldsymbol{x}_{-m:i-1})$.
7:     **else**
8:         Generate the permutation of token set $\theta_i = h(k, \boldsymbol{s}_i)$.
9:         Calculate DiP-reweight $P_W(\cdot|\boldsymbol{x}_{-m:i-1}, \theta_i)$ with $\alpha$ and Def. 4.1.
10:         Sample the next token $x_i$ using distribution $P_W(\cdot|\boldsymbol{x}_{-m:i-1}, h(k, \boldsymbol{s}_i))$.
11: **return** $\boldsymbol{x}_{1:n}$.

---

Leveraging the inherent symmetry of permutations, we can demonstrate that a weighted combination of $\alpha$-reweight and $(1 - \alpha)$-reweight yields a distribution-preserving reweight strategy. It is pivotal to recognize that both $\alpha$-reweight and $(1 - \alpha)$-reweight amplify the sum of the probability of green-list tokens. Therefore, the combined effect of these reweight functions still exhibits a preference for the green list tokens. The formal articulation of our distribution-preserving reweight strategy is presented subsequently.

**Definition 4.1** (DiP-reweight). Given the original token distribution $P_M(t|\boldsymbol{x}), \forall t \in V$, where $\boldsymbol{x} \in \Sigma$ is the previous token sequence. Denote by $\theta = \{t_1, ..., t_N\}$ the cipher, which is a permutation of $V$. Let

$$F(i|\theta) := \max \left\{ \sum_{j=1}^{i} P_M(t_j|\boldsymbol{x}) - \alpha, 0 \right\} + \max \left\{ \sum_{j=1}^{i} P_M(t_j|\boldsymbol{x}) - (1 - \alpha), 0 \right\}, \; i \in [1, N], \; (3)$$

where $0 \leq \alpha \leq 1$. The DiP-reweight strategy can be represented by $P_W(t_i|\boldsymbol{x}, \theta) := F(i|\theta) - F(i - 1|\theta)$ and $F(0|\theta) := 0$.

It is easy to show that $P_W(t_i|\boldsymbol{x}, \theta)$ is a distribution on $V$. Firstly, as $F(i|\theta)$ is monotonously increasing with $i$, we have $P_W(t_i|\boldsymbol{x}, \theta) = F(i|\theta) - F(i - 1|\theta) \geq 0$. Secondly, the sum of the probability of all tokens is $\sum_{i=1}^{N} P_W(t_i|\boldsymbol{x}, \theta) = \sum_{i=1}^{N} (F(i|\theta) - F(i - 1|\theta)) = F(N|\theta) = 1 - \alpha + \alpha = 1$.

**Theorem 4.2.** *DiP-reweight is a distribution-preserving reweight strategy.*

We defer the proof of Theorem 4.2 to Appendix D. With the DiP-reweight approach, the generation of *i.i.d.* ciphers, denoted as $\theta_i$, becomes essential for crafting a distribution-preserving watermark. Let $k$ signify a stochastic secret key derived from the key space $K$ following the distribution $P_K$, let $\boldsymbol{s} \in \mathcal{V}$ be a texture key, which is a sub-sequence of the previously generated context. Considering the context generated prior to time step $t$ denoted as $\boldsymbol{x}_{1:t-1}$, instances of texture keys encompass $x_{t-1}, \boldsymbol{x}_{t-3:t-1}$, and $\boldsymbol{x}_{1:t-1}$. We introduce a hash function, $h(k, \boldsymbol{s}) : K \times \mathcal{V} \to \Theta$, orchestrating the mapping of a secret key in conjunction with a texture key. $\boldsymbol{s} \in \mathcal{V}$ to a permutation of the token set $V$. The chosen hash function $h$ must adhere to the following conditions: a) For distinct texture keys, i.e., $\boldsymbol{s}_1 \neq \boldsymbol{s}_2$, $h(k, \boldsymbol{s}_1)$ ought to be statistically independent from $h(k, \boldsymbol{s}_2)$ when $k \sim P_K(k)$, and b) Upon holding $\boldsymbol{s}$ constant, every $V_i^p \in \Sigma$ should exhibit a uniform likelihood of being selected given a random key, specifically, $\forall V_i^p \in \Sigma, \mathbb{E}_{k \sim P_K}[\mathbf{1}_{h(k,\boldsymbol{s})=V_i^p}] = 1/N!$.

There exists hash functions meeting the aforementioned criteria, one example being the function delineated in Kirchenbauer et al. (2023). Under such conditions, the cipher $\theta_i$ can be deemed *i.i.d.* if the texture key $\boldsymbol{s}_i$ is distinctive for each instance. To ensure this uniqueness, a register is employed to retain texture keys generated in prior steps. If a recurring texture key is identified in the historical log, the inherent distribution is utilized for the subsequent token generation. Otherwise, the DiP-reweight is applied for the token generation. The detailed methodology is delineated in Alg. 1.

**Corollary 4.3.** *DiPmark (Alg. 1) is a distribution-preserving watermark.*

This can be easily validated by combining the distribution-preserving property of DiP-reweight and the independence of cipher $\theta_i$.

---

**Algorithm 2** DiPmark detector

---

1: **Input:** text $\boldsymbol{x}_{1:n}$, watermark key $k$, volume of the token set $N$, permutation generation function $h$, green list separator $\gamma$, threshold $z$.
2: Initialize the green token indexer of $\gamma$ with 0: $L_G(\gamma) = 0$.
3: **for** $i = 2, ..., n$ **do**
4:      Generate a texture key $\boldsymbol{s}_i$ based on $\boldsymbol{x}_{1:i-1}$.
5:      Generate the permutation of token set $\theta_i = h(k, \boldsymbol{s}_i)$.
6:      Calculate the list of green tokens via $G = \theta_i[\lceil \gamma N \rceil : N]$.
7:      **if** $x_i \in G$ **then**
8:          Increase the number of green token indexer by 1: $L_G(\gamma) = L_G(\gamma) + 1$.
9: Calculate the score: $\Phi(\gamma, \boldsymbol{x}_{1:n}) = \frac{L_G(\gamma) - (1-\gamma)n}{\sqrt{n}}$.
10: **return** $\Phi(\gamma, \boldsymbol{x}_{1:n}) > z$.

---

## 5   WATERMARK DETECTION

Given that DiPmark is founded on the principle of promoting the utilization of green-list tokens, we can leverage a hypothesis test to identify the presence of DiPmark. In the context of a predetermined red-green list separator $\gamma \in [0, 1]$, we classify the initial $\lceil \gamma N \rceil$ tokens within the token set permutation as belonging to the red list, while the remaining tokens are categorized as part of the green list. Given a text sequence $\boldsymbol{x}_{1:n}$, we establish the null hypothesis $H_0$: $\boldsymbol{x}_{1:n}$ is generated without any awareness of the red-green list rule. Here, $L_G(\gamma)$ denotes the count of green tokens within $\boldsymbol{x}_{1:n}$ in relation to $\gamma$. Under the null hypothesis $H_0$, $L_G(\gamma)$ follows a binomial distribution with parameters $p = (1 - \gamma)$ and $n$ total trials. We introduce $\Phi(\gamma, \boldsymbol{x}_{1:n}) := (L_G(\gamma) - (1 - \gamma)n)/\sqrt{n}$, and by employing Hoeffding's inequality, we derive the subsequent concentration bound:

$$\Pr(\Phi(\gamma, \boldsymbol{x}_{1:n}) \geq t) \leq e^{-2t^2}, \quad \forall t \in \mathbb{R}.$$

We proceed to reject the null hypothesis and detect the watermark if $\Phi(\gamma, \boldsymbol{x}_{1:n})$ surpasses a predefined threshold. For instance, setting the threshold as $\Phi(\gamma, \boldsymbol{x}_{1:n}) \geq 2$ results in rejecting $H_0$ (indicating watermark presence) while maintaining a false positive rate below $e^{-8}$.

It is noteworthy that we diverge from Kirchenbauer et al. (2023) by avoiding the use of the z-test statistic $(L_G(\gamma) - (1 - \gamma)n)/\sqrt{n\gamma(1 - \gamma)}$. The z-test assumes a Gaussian distribution for the test statistic, but in cases of small text sequence lengths $n$, the approximation becomes imprecise, potentially leading to inaccurate estimation of false positive rates.

As DiPmark can always elevate the sum of the probability of the green list tokens with arbitrary $\gamma$, selecting a suitable $\gamma$ becomes a significant consideration. To enable fair comparison with the Soft watermark in Kirchenbauer et al. (2023), we fix $\gamma = 0.5$ for both watermarks during the detection phrase, which means the first half of the permutation set is the red list and the second half is the green list.

**Green token ratio.** The term $L_G(\gamma)/n$ signifies the proportion of green tokens within a sequence of tokens, while $1 - \gamma$ denotes the expected green token proportion in an unwatermarked sequence. Our metric, referred to as the "green token ratio", is defined as the difference between these two values, expressed as $L_G(\gamma)/n - (1 - \gamma)$. This ratio quantifies the bias towards green tokens within the text sequence. It is worth noting that $\Phi(\gamma, \boldsymbol{x}_{1:n})$ scales proportionally with the square root of $n$ times the green token ratio. With a fixed green token ratio, detecting longer sequences using $\Phi(\gamma, \boldsymbol{x}_{1:n})$ becomes more straightforward because they will show a lower false positive rate. The validity of this analysis is also confirmed in Section 6.4.

**Detecting efficiency comparison.** Similar to the detection algorithms presented in Kirchenbauer et al. (2023) and Christ et al. (2023), our watermark detection process is highly efficient, requiring only a single pass through the provided text sequence. However, it is worth noting that the detection algorithm outlined in Kuditipudi et al. (2023) necessitates iterating through the sequence a staggering 5000 times, which is notably inefficient when compared to our approach. In terms of computational costs, our detection algorithm proves to be significantly faster. For instance, in a simulation comparing the two methods, our algorithm only requires 90 seconds to detect 1,000 watermarked sequences generated by LLaMA-2, whereas the algorithm in Kuditipudi et al. (2023) demands at least 10 hours for the same task.

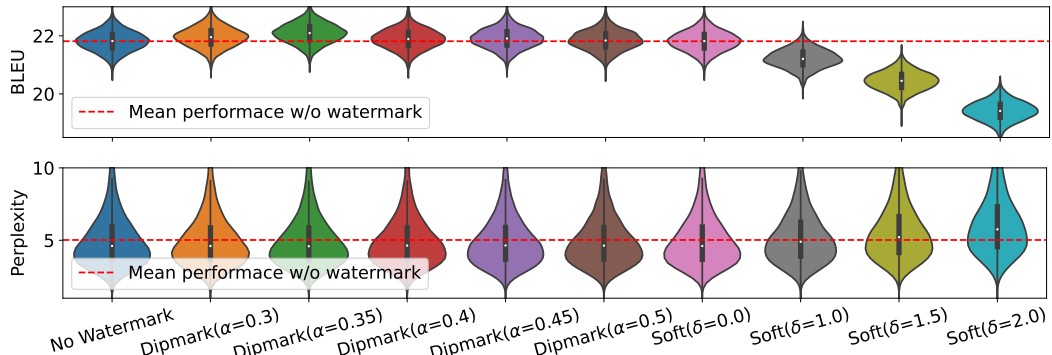

**Figure 1:** Empirical verification of distribution-preserving property of DiPmark. **Top:** Violin plot of Machine Translation BLEU. **Bottom:** Violin plot of Text Summarization Perplexity. We can see the Soft watermarks significantly degrade the text quality, while DiPmarks preserve the text quality.

**Table 2:** Performance of different watermarking methods on machine translation and text summarization. We use F1 scores of BERTScore and scale BERTScore and ROUGE-1 with a factor of 100.

| | Machine Translation | | Text Summarization | | |
|---|---|---|---|---|---|
| | BERTScore↑ | BLEU↑ | BERTScore↑ | Perplexity↓ | ROUGH-1↑ |
| **No Watermark** | 0.559±0.003 | 21.8±0.3 | 0.3273±0.0008 | 5.021±0.018 | 0.3855±0.0009 |
| **Soft ($\delta$=0.0)** | 0.560±0.003 | 21.8±0.3 | 0.3273±0.0008 | 5.021±0.018 | 0.3855±0.0009 |
| **Soft ($\delta$=1.0)** | 0.557±0.003 | 21.2±0.3 | 0.3237±0.0008 | 5.309±0.019 | 0.3816±0.0009 |
| **Soft ($\delta$=1.5)** | 0.550±0.003 | 20.4±0.3 | 0.3209±0.0008 | 5.660±0.021 | 0.3793±0.0009 |
| **Soft ($\delta$=2.0)** | 0.539±0.003 | 19.4±0.3 | 0.3146±0.0008 | 6.241±0.023 | 0.3725±0.0009 |
| **Kuditipudi et al. (2023)** | 0.560±0.003 | 21.7±0.3 | 0.3270±0.0008 | 5.021±0.021 | 0.3854±0.0009 |
| **Hu et al. (2023)** | 0.563±0.003 | 21.8±0.3 | 0.3271±0.0008 | 5.023±0.018 | 0.3857±0.0009 |
| **DiPmark ($\alpha$=0.3)** | 0.561±0.003 | 22.0±0.3 | 0.3279±0.0008 | 5.014±0.018 | 0.3861±0.0009 |
| **DiPmark ($\alpha$=0.35)** | 0.562±0.003 | 22.1±0.3 | 0.3274±0.0008 | 4.998±0.018 | 0.3856±0.0009 |
| **DiPmark ($\alpha$=0.4)** | 0.561±0.003 | 21.9±0.3 | 0.3277±0.0008 | 5.001±0.018 | 0.3862±0.0009 |
| **DiPmark ($\alpha$=0.45)** | 0.562±0.003 | 21.9±0.3 | 0.3269±0.0008 | 5.024±0.018 | 0.3852±0.0009 |
| **DiPmark ($\alpha$=0.5)** | 0.562±0.003 | 21.8±0.3 | 0.3272±0.0008 | 5.014±0.018 | 0.3859±0.0009 |

**Resilience analysis.** Our detecting algorithm is **provably** robust against text modification attacks. Given a watermarked sentence $x_{1:n}$, we use the preceding $a$ tokens to serve as the texture key to generate the token permutation. Denote by $\epsilon$ the portion of text modifications and $z$ the threshold of hypothesis testing, we have $\forall \epsilon \leq (\Phi(\gamma, x_{1:n}) - z)/((a+1)\sqrt{n})$. That is, the watermarked sequence under attack will still be detected as watermarked. We defer the detailed analysis in Appendix B.

## 6 EXPERIMENTS

Our experimental section consists of four parts. In the first three parts, we comparing the stealthy, efficient, and resilient of DiPmark with the SOTA watermark methods (Kirchenbauer et al., 2023; Kuditipudi et al., 2023; Hu et al., 2023). Notice, as (Christ et al., 2023) is a pure theoretical paper without any experiments and code, we are not able to compare our work with it. In the fourth part, we compare the detectability of DiPmark with the Soft watermark introduced in Kirchenbauer et al. (2023) on a LLaMA-2 (chat, 7B) model (Touvron et al., 2023). A detailed experimental setting is in Appendix E.

**General experimental observation.** We find that our DiPmark, configured with $\alpha$ values of 0.45 and 0.5, exhibits comparable levels of detectability and robustness comparing with the Soft watermark with $\delta = 1.5$. Importantly, our DiPmark maintains the same level of text quality as the original model, owing to its inherent distribution-preserving property.

### 6.1 STEALTHINESS

A watermark is stealthy if and only if it is distribution-preserving, we will empirically varify the distribution-preserving property of different watermarks. Since DiPmark is **provably** distribution-preserving (Corollary 4.3), we only use this experiment as an additional support for this property.

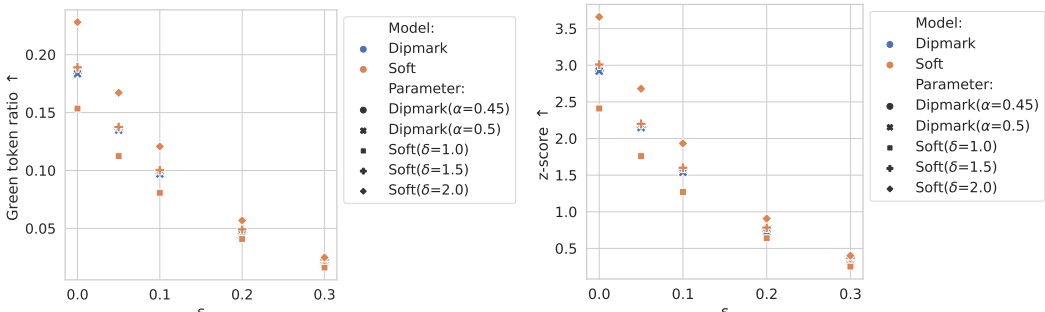

**Figure 2:** Robustness evaluation of DiPmark on poetry generation task. **Left.** Average green token ratio w.r.t. portion of perturbation $\epsilon$. **Right.** Average z-score w.r.t. portion of perturbation $\epsilon$.

**Table 4:** AUC score of different watermarks under varying attack strength $\epsilon$ on poetry generation task. Each row is evaluated over around 300 watermarked and 500 non-watermarked sequences of length n = 260 ± 5.

| AUC | Random text modification | | | | Paraphrasing attack | | | |
|---|---|---|---|---|---|---|---|---|
| | $\epsilon = 0.0$ | $\epsilon = 0.1$ | $\epsilon = 0.2$ | $\epsilon = 0.3$ | $\epsilon = 0.0$ | $\epsilon = 0.1$ | $\epsilon = 0.2$ | $\epsilon = 0.3$ |
| Soft ($\delta$=1.5) | **0.9990** | **0.9883** | **0.9521** | 0.8033 | **0.9990** | **0.9894** | 0.9469 | 0.8157 |
| Kuditipudi et al. (2023) | 0.9951 | 0.9461 | 0.8979 | 0.7815 | 0.9951 | 0.9529. | 0.9013 | 0.7711 |
| Hu et al. (2023) | 0.9936 | 0.9297 | 0.8391 | 0.7574 | 0.9936 | 0.9368 | 0.8325 | 0.7661 |
| DiPmark ($\alpha$=0.45) | **0.9990** | 0.9859 | 0.9515 | **0.8060** | **0.9990** | 0.9871 | **0.9503** | **0.8216** |

**Settings.** We follow the evaluation process of Hu et al. (2023), where we assess the performance of DiPmark with two seq2seq models: machine translation (MT) and text summarization (TS). For the TS task, our experimentation employs the BART-large model (Liu et al., 2020) in conjunction with the CNN-DM corpus (Hermann et al., 2015) as our designated testing dataset. The MT task, on the other hand, revolves around English-to-Romanian translation. For this purpose, we employ the Multilingual BART (MBart) model (Liu et al., 2020) on the WMT'14 En-Ro corpus. Specifically for DiPmark, we select values for $\alpha$ from the set $\{0.3, 0.35, 0.4, 0.45, 0.5\}$, while for the Soft watermark (Kirchenbauer et al., 2023), we choose green list bias values $\delta$ from the set $\{0.0, 1.0, 1.5, 2.0\}$ alongside a fixed green list separator $\gamma = 0.5$, indicating that 50% of tokens are green while the remainder are red. It is important to note that the Soft watermark with $\delta = 0.0$ is essentially equivalent to no watermark since it does not promote the probability of green list tokens.

**Result analysis.** Upon examining Figure 1 and the data presented in Table 2, a striking pattern emerges. Across all $\alpha$ values in the range $\{0.3, 0.35, 0.4, 0.45, 0.5\}$, the BLEU scores in the machine translation tasks and the perplexity values in the text summarization tasks remain consistently similar between DiPmark and the original language model. However, as we increase the $\delta$ values in the Soft watermark, a notable degradation in text quality becomes evident. A more comprehensive set of results is provided in Appendix F.1.

## 6.2 EFFICIENCY

We compare the time for detecting 1 and 1,000 watermarked text with different detection algorithm. The task is poetry generation with LLaMA-2 (chat, 7B). We use the same GPU (NVIDIA A6000) for all experiments. From Table 3 we see the detecting algorithms of Soft and DiPmark are efficient without accessing LMs, while Hu et al. (2023) requires additional access to LMs and Kuditipudi et al. (2023) needs significantly longer time.

**Table 3:** Comparison of detecting time for different watermarks.

| Number of samples | 1 | 1,000 |
|---|---|---|
| Soft (Kirchenbauer et al., 2023) | 0.3s | 92s |
| Kuditipudi et al. (2023) | 80s | 12h |
| Hu et al. (2023) (need LM access) | 3.4s | 412s |
| DiPmark | 0.3s | 90s |

## 6.3 RESILIENCE

We conduct experiments to test the resiliency of the our DiPmark and the Soft watermark in Kirchenbauer et al. (2023). In this context, we use the poetry generation tasks with 1,000 generated sequences on LLaMA-2. For resilience evaluation, we manipulating about $\epsilon \in \{0.05, 0.1, 0.2, 0.3\}$ portion of the text tokens through random text modifications and paraphrasing attacks.

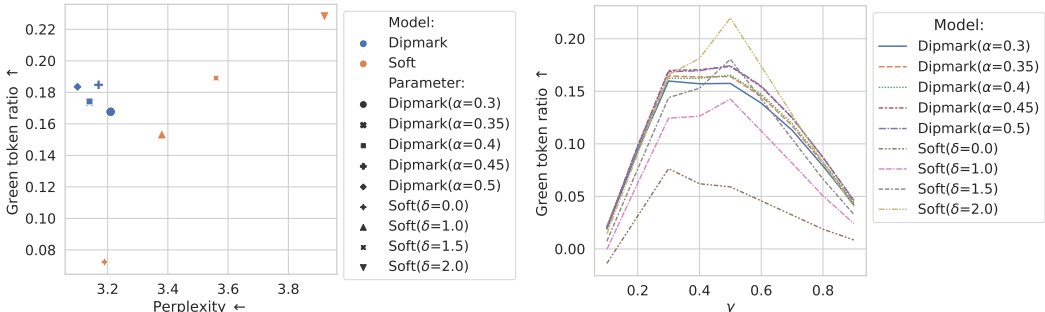

**Figure 3: Left:** Average perplexity vs green token rate with $\gamma = 0.5$ on poetry generation tasks. **Right:** Average green token ratio with different $\gamma$.

**Table 5:** Empirical error rates for watermark detection on poetry generation. Each row is averaged over around 300 watermarked and 500 non-watermarked sequences of length $n = 260 \pm 5$. We select the threshold $z = 1.85$ (false positive rate 0.001) and $z = 2$ (false positive rate 0.00034).

| | $z = 1.85$ | | | | $z = 2$ | | | |
|---|---|---|---|---|---|---|---|---|
| | FPR↓ | TNR↑ | TPR↑ | FNR↓ | FPR↓ | TNR↑ | TPR↑ | FNR↓ |
| **Soft ($\delta$=1.0)** | 0.0060 | 0.9940 | 0.7314 | 0.2686 | 0.0 | 1.0 | 0.5714 | 0.4286 |
| **Soft ($\delta$=1.5)** | 0.0060 | 0.9940 | 0.9204 | 0.0796 | 0.0 | 1.0 | 0.8134 | 0.1867 |
| **Soft ($\delta$=2.0)** | 0.0060 | 0.9940 | 0.9829 | 0.0171 | 0.0 | 1.0 | 0.9562 | 0.0439 |
| **DiPmark ($\alpha$=0.45)** | 0.0060 | 0.9940 | 0.9167 | 0.0833 | 0.0 | 1.0 | 0.8095 | 0.1905 |
| **DiPmark ($\alpha$=0.5)** | 0.0060 | 0.9940 | 0.9176 | 0.0824 | 0.0 | 1.0 | 0.7802 | 0.2198 |

**Result Analysis.** Figure 2 elucidates the evolution of the average green token ratio and the average z-score concerning the attack strength parameter $\epsilon$. Notably, both metrics exhibit a diminishing trend as $\epsilon$ increases. In Table 4, we report the AUC score of different watermarks under varying attack strength $\epsilon$. The analysis underscores that, when $\epsilon$ remains below 0.3, DiPmark demonstrates robust performance in effectively detecting watermarked sentences.

## 6.4 DETECTABILITY

**Settings.** We evaluate the detectability of our watermark on poetry generation and text summarization tasks using LLaMA-2. We generate 1,000 examples for each tasks. We also select $\alpha \in \{0.3, 0.35, 0.4, 0.45, 0.5\}$ for DiPmark, and $\delta \in \{0.0, 1.0, 1.5, 2.0\}$ and $\gamma = 0.5$ for Soft watermark (Kirchenbauer et al., 2023). During detection, we also use $\gamma = 0.5$. We report the green token ratio (defined in 5), the score of $\Phi(\gamma, \boldsymbol{x})$ (z-score), and the Type I (FPR) and II (FNR) errors.

**Result analysis.** The results for poetry generation are visually depicted in Figure 3 and Table 5. Broadly speaking, our DiPmark variants with $\alpha = 0.45$ and 0.5 exhibit performance comparable to that of the Soft watermark with $\delta = 1.5$, where $\delta = 1.5$ corresponds to an augmentation of 1.5 to the green token logits. In Figure 3 (left), it is evident that our DiPmark variants with $\alpha = 0.45$ and 0.5 yield green token ratios akin to those of the Soft watermark with $\delta = 1.5$ without any discernible degradation in text quality. Figure 3 (right) delves into the impact of different green list separators $\gamma$, revealing that, for most watermark models, $\gamma = 0.5$ yields the highest green token ratio, underscoring its suitability as a reasonable choice for watermark detection. The empirical error rates for watermark detection in poetry generation are reported in Table 5, showcasing the commendable performance of our DiPmark with low false positive rates while maintaining a high true positive rate. For the text summarization results, please refer to Appendix F.2.

## 7 CONCLUSION

In summary, we present DiPmark, a novel watermarking solution tailored for LLMs. Our meticulously designed DiPmark exhibits the crucial attributes of stealthiness, efficiency, and resilience, attributes that we rigorously substantiate through a combination of theoretical analyses and empirical investigations. This innovation represents a significant step forward in LLM watermarking, ensuring both the security and quality of machine-generated content. Importantly, our work not only strengthens the theoretical foundations but also imparts practical insights that are valuable for the industrial deployment of LLM watermarking technologies.

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

## A    LIMITATION AND FUTURE WORK

A prominent limitation of DiPmark lies in its detectability, which is akin to the Soft watermark at $\delta = 1.5$. This limitation may render it unsuitable for scenarios necessitating both high true negative rate and high true positive rate.

Future endeavors should focus on enhancing the detectability of distribution-preserving watermarks. This could be realized by assigning greater weight to the green-list tokens during the watermarking process. Additionally, a promising avenue for exploration involves the design of a more robust distribution-preserving watermark, potentially through the integration of multiple detectors. These directions represent promising opportunities for advancing the efficacy and applicability of watermarking techniques on large language models.

## B    DIPMARK IS PROVABLY ROBUST TO TEXT MODIFICATION

In this section, we establish that our DiPmark possesses provable robustness against textual modification attacks, encompassing text insertion, substitution, and deletion.

**Problem formulation.** Let $\boldsymbol{x}_{1:n}$ represent a watermarked sentence. To generate the cipher $\theta$ at the i-th iteration, we employ a hash function $h$, a confidential key $k$, and a texture key $\boldsymbol{s} := \boldsymbol{x}_{i-a:i-1}, a \geq 1$. This indicates that the preceding $a$ tokens serve as the texture key for the watermarking of the token situated at position $i$. During the detection phase, the formula $\Phi(\gamma, \boldsymbol{x}_{1:n}) := (L_G(\gamma) - (1 - \gamma)n)/\sqrt{n}$ coupled with a threshold $z$ is applied to ascertain if the text has been watermarked. Notably, within $\Phi(\gamma, \boldsymbol{x}_{1:n})$, the sole variable associated with textual modification assaults is $L_G(\gamma)$. Consequently, our primary objective is to discern the most severe reduction in $L_G(\gamma)$ for a single token alteration.

**Worst-case perturbation analysis.** Supposing the token $x_i$ in $\boldsymbol{x}_{1:n}$ undergoes modification, this will instigate a reduction in $L_G(\gamma)$ through two channels: a) Initially, the token $x_i$ may be categorized as a green token, but post-alteration, it either gets eliminated or transitions into a red token, leading to a potential decline in the number of green tokens $L_G(\gamma)$ by at most 1. b) Since the list of red-green tokens for $x_{i+1}, ..., x_{i+a}$ is generated by hashing the token $x_i$, its subsequent alteration could cause $x_{i+1}, ..., x_{i+a}$ to turn into red tokens. In this scenario, the number of green tokens $L_G(\gamma)$ may shrink by a maximum of $a$. As a result, the greatest decline in $L_G(\gamma)$ for a single token modification stands at $a + 1$.

**Determination of certified radius.** Let $\epsilon \in [0, 1]$ denote the fraction of altered tokens (termed as the attack budget). We aim to illustrate a certified radius $\epsilon_0$, where, for all perturbations confined within the budget $\epsilon \leq \epsilon_0$, the altered watermarked sequence continues to be recognized as watermarked.

**Theorem B.1.** *Given $\Phi(\gamma, \boldsymbol{x}_{1:n}) := (L_G(\gamma) - (1 - \gamma)n)/\sqrt{n}$ and a threshold $z$, the certified radius of the watermarked sequence $\boldsymbol{x}_{1:n}$ is*

$$\epsilon_0 = \frac{1}{n}\left(\frac{\sqrt{z^2 + 4AC} - z}{2A}\right)^2 - 1,$$

*where $A = 2 + a - \gamma, C = L_G(\gamma) + (a + 1)n$.*

*Proof.* Notice based on above discussion, the worst-case decrease on $L_G(\gamma)$ per token modification is $a + 1$. If we are allowed to perturbed $\epsilon$ portion of the text, the worst-case decrease on $L_G(\gamma)$ will be $(a + 1)\epsilon n$. Denoted by $\boldsymbol{x}_{1:n'}$ the perturbed text. Assume we can still correctly detect the watermarked sequence, which means

$$(L_G(\gamma) - (1 - \gamma)n' - (a + 1)\epsilon n)/\sqrt{n'} \geq z.$$

Notice, the left hand side of the above equation is decreasing with $n'$, as we perturbed $\epsilon$ portion of the text, the maximum of the possible $n'$ is $n' = (1 + \epsilon)n$, i.e., all modifications are text insertion. In this case, we need to solve

$$(L_G(\gamma) - (1 - \gamma)(1 + \epsilon)n - (a + 1)\epsilon n)/\sqrt{(1 + \epsilon)n} \geq z.$$

we have

$$\epsilon \leq \frac{1}{n} \left( \frac{\sqrt{z^2 + 4AC} - z}{2A} \right)^2 - 1,$$

where $A = 2 + a - \gamma$ and $C = L_G(\gamma) + (a+1)n$.

Therefore, for any text modification with budget $\epsilon \leq \frac{1}{n}(\frac{\sqrt{z^2+4AC}-z}{2A})^2 - 1$, our algorithm can still detect the watermarked sequence. $\square$

In the following theorem, we provide a more simple certified radius assuming the text length is not changed by perturbations.

**Theorem B.2.** *Assuming the sequence length $n$ is not changed through text modifications. Given $\Phi(\gamma, \boldsymbol{x}_{1:n}) := (L_G(\gamma) - (1 - \gamma)n)/\sqrt{n}$ and a threshold $z$, the certified radius of the watermarked sequence $\boldsymbol{x}_{1:n}$ is $\epsilon_0 = \frac{\Phi(\gamma, \boldsymbol{x}_{1:n}) - z}{(a+1)\sqrt{n}}$.*

*Proof.* Notice based on above discussion, the worst-case decrease on $L_G(\gamma)$ per token modification is $a + 1$. If we are allowed to perturbed $\epsilon$ portion of the text, the worst-case decrease on $L_G(\gamma)$ will be $(a + 1)\epsilon n$. Assume we can still correctly detect the watermarked sequence, which means

$$(L_G(\gamma) - (1 - \gamma)n - (a + 1)\epsilon n)/\sqrt{n} \geq z,$$

we have $\epsilon \leq \frac{\Phi(\gamma, \boldsymbol{x}_{1:n}) - z}{(a+1)\sqrt{n}}$. Therefore, for any text modification with budget $\epsilon \leq \frac{\Phi(\gamma, \boldsymbol{x}_{1:n}) - z}{(a+1)\sqrt{n}}$, our algorithm can still detect the watermarked sequence. $\square$

## C  RELATED WORK

**Reweight-based watermarking framework.** In a recent seminal work, Kirchenbauer et al. (2023) introduced a pioneering watermarking scheme tailored for LLMs, backed by formal guarantees. Their work demonstrated that watermark embedding could be accomplished by altering the token distribution during generation, targeting outputs with substantial entropy. However, this approach inevitably leads to a pivotal change in the distribution of the generated text, potentially compromising the quality of the generated content.

To maintain an unaltered output distribution in watermarked content, alternative strategies have been explored. Christ et al. (2023) and Kuditipudi et al. (2023) employed the inverse sampling method to generate watermarked token distributions. Notably, Christ et al. (2023)'s method faces resilience issues under modifications and lacks empirical validation for detectability. Meanwhile, Kuditipudi et al. (2023)'s approach necessitates the secret key distribution during detection, potentially compromising data security and watermark stealthiness. Moreover, their detection process involves hundreds of resampling steps from the secret key distribution, which is inefficient for lengthy texts. Hu et al. (2023) used inverse sampling and permutation based reweight methods for watermarking, but the detector requires access of the language model API, undermining its operational efficiency. Aaronson's ongoing watermarking project (Aaronson, 2022) employs n-gram hashing for reweighting the next-token distribution, though specific details are currently unavailable.

The landscape also includes several schemes (Abdelnabi & Fritz, 2021; Qiang et al., 2023; Yoo et al., 2023; Munyer & Zhong, 2023) that incorporate an ML model within the watermarking algorithm itself. However, these constructions lack formal assurances and rely on heuristic arguments for satisfying the criteria of Stealthiness, Efficiency, and Resilience.

Our research aligns closely with the findings presented in Kirchenbauer et al. (2023). In their methodology, they employed watermarking for text derived from a language model by bifurcating the token set into designated 'red' and 'green' lists. The division is determined by a random seed that is contingent on the secret key coupled with a hash of priorly generated tokens. The authors accentuated the prominence of green tokens during the sampling phase by reweighting the token log-probabilities. Building on this foundation, our research retains the red-green list configuration, but introduces an evolved family of permutation-based reweight strategies. This dual approach ensures: 1) a promoted utilization of green tokens, and 2) equivalency in distribution between a sample from the watermarked language model and one from the original language model.

**Post-hoc detectors.** Post-hoc detection stands as a notable alternative to watermarking, focusing on the retrospective analysis of machine-generated text. This could be achieved through leveraging features inherent to language models or by refining pre-existing, expansive language models to function as detectors, as elaborated by (Zellers et al., 2019). Notably, specific implementation nuances, such as sampling methodologies, can be discerned through reverse engineering the generated text, a process detailed by (Tay et al., 2020). There are also post-hoc detectors designed for the modern large language models (Mitchell et al., 2023; Tian, 2023; Kirchner et al., 2023), which are models specifically trained for the binary detection task. However, there is a growing sentiment that those detection methodologies are diminishing in efficacy in tandem with the evolution of language model capabilities. As Gambini et al. (2022) observed, detection mechanisms that were adept with GPT-2 have encountered challenges with GPT-3. Besides, the text rephrasing model in (Krishna et al., 2023) bypassing prevalent post-hoc detectors like GPTZero (Tian, 2023), DetectGPT (Mitchell et al., 2023), and OpenAI's proprietary detector (Kirchner et al., 2023). Additionally, a pertinent observation made by Chakraborty et al. (2023) suggests that as AI-generated content becomes increasingly indistinguishable from human-produced text, the demands on post-hoc detectors to analyze more extended text segments will escalate.

**Steganography.** Steganography involves embedding concealed messages in channels such as natural language or images, ensuring only intended recipients can discern the message while others remain unaware (Hopper et al., 2002). When applied to watermarking, the aim is stealthy. Yet, known steganography techniques might not achieve this without certain entropy-related assumptions. In scenarios where language model prompts can be chosen adversarially, the need for stealthy persists. This discrepancy arises due to differences in access levels that watermarking and steganography have to the model's output distribution. In steganography, there's only oracle access to this distribution. Conversely, our watermarking approach gets a detailed view of the token's probability distribution. Hence, while steganography either relies on entropy assumptions (Hopper et al., 2002) or compromises security with low entropy channels (Dedić et al., 2009), our watermark remains stealthy irrespective of the text's entropy. This is achieved by leveraging the full distribution access and using it as a foundation for embedding watermarks. Kaptchuk et al. (2021) offers encoding similar access. However, it presupposes equal decoding access, which is impractical for watermarking as the detection algorithm won't typically have the initiating prompt, thus remaining ignorant of the distribution.

## D MISSING PROOFS

### D.1 PROOF OF THEOREM 4.2

*Proof.* We need to show $\forall t \in V, \mathbb{E}_\theta[P_W(t|\boldsymbol{x}, \theta)] = P_M(t|\boldsymbol{x})$. Recall $\theta$ is uniformly distributed on $\Theta$, we have

$$
\begin{aligned}
\mathbb{E}_{\theta \sim P_\Theta}[P_W(t|\boldsymbol{x}, \theta)] &= \sum_{V^p \in \Theta} \mathbb{E}_{\theta \sim P_\Theta}[P_W(t|\boldsymbol{x}, V^p)\mathbf{1}_{\theta=V^p}] \\
&= \sum_{V^p \in \Theta} [P_W(t|\boldsymbol{x}, V^p)]\mathbb{E}_{\theta \sim P_\Theta}[\mathbf{1}_{\theta=V^p}] \\
&= \frac{1}{N!} \sum_{V^p \in \Theta} P_W(t|\boldsymbol{x}, V^p).
\end{aligned}
\tag{4}
$$

Given an token $t$ and a permutation of the token list $V^p$, denote by $E_{V^p}(t)$ the position of $t$ in the ordered token set $V^p$. Let $V^{p^r}$ be the reversed permutation of $V^p$, notice $t$ is the $(N+1-E_{V^p}(t))$-th element in $V^{p^r}$. Given an arbitrary permutation pair $(V^p, V^{p^r})$, $V^p := \{t_1, ..., t_N\}$. We will show

$$
P_W(t|\boldsymbol{x}, V^p) + P_W(t|\boldsymbol{x}, V^{p^r}) = 2P_M(t|\boldsymbol{x}).
$$

For the ease of notation we denote by $i = E_{V^p}(t)$, we have $t_i = t$. From the definition of DiP-reweight we know $P_W(t|\boldsymbol{x}, V^p) = F(E_{V^p}(t)|V^p) - F(E_{V^p}(t)-1|V^p) = F(i|V^p) - F(i-1|V^p)$. So we need to show

$$
F(i|V^p) - F(i-1|V^p) + F(N+1-i|V^{p^r}) - F(N-i|V^{p^r}) = 2P_M(t|\boldsymbol{x}).
$$

As $\sum_{j=1}^{N} P_M(t_j|\boldsymbol{x}) = 1$, we have

$$F(N+1-i|V^{p^r}) = \max\left\{\sum_{j=1}^{N+1-i} P_M(t_{N+1-j}|\boldsymbol{x}) - \alpha, 0\right\} + \max\left\{\sum_{j=1}^{N+1-i} P_M(t_{N+1-j}|\boldsymbol{x}) - (1-\alpha), 0\right\}$$

$$= \max\left\{\sum_{j=i}^{N} P_M(t_j|\boldsymbol{x}) - \alpha, 0\right\} + \max\left\{\sum_{j=i}^{N} P_M(t_j|\boldsymbol{x}) - (1-\alpha), 0\right\}$$

$$= \max\left\{(1-\alpha) - \sum_{j=i}^{i-1} P_M(t_j|\boldsymbol{x}), 0\right\} + \max\left\{\alpha - \sum_{j=1}^{i-1} P_M(t_j|\boldsymbol{x}), 0\right\},$$

$$(5)$$

and

$$F(i-1|V^p) = \max\left\{\sum_{j=1}^{i-1} P_M(t_j|\boldsymbol{x}) - \alpha, 0\right\} + \max\left\{\sum_{j=1}^{i-1} P_M(t_j|\boldsymbol{x}) - (1-\alpha), 0\right\}. \quad (6)$$

By $(\max\{A, 0\} - \max\{-A, 0\}) = A, \forall A \in \mathbb{R}$, we have

$$F(N+1-i|V^{p^r}) - F(i-1|V^p) = (1-\alpha) - \sum_{j=i}^{i-1} P_M(t_j|\boldsymbol{x}) + \alpha - \sum_{j=1}^{i-1} P_M(t_j|\boldsymbol{x})$$

$$(7)$$

$$= 1 - 2\sum_{j=i}^{i-1} P_M(t_j|\boldsymbol{x}).$$

Analogously, we have

$$F(N-i|V^{p^r}) - F(i|V^p) = 1 - 2\sum_{j=i}^{i} P_M(t_j|\boldsymbol{x}). \quad (8)$$

Thus,

$$P_W(t|\boldsymbol{x}, V^p) + P_W(t|\boldsymbol{x}, V^{p^r}) = F(i|V^p) - F(i-1|V^p) + F(N+1-i|V^{p^r}) - F(N-i|V^{p^r})$$

$$= (1 - 2\sum_{j=i}^{i-1} P_M(t_j|\boldsymbol{x})) - (1 - 2\sum_{j=i}^{i} P_M(t_j|\boldsymbol{x}))$$

$$= 2P_M(t_i|\boldsymbol{x}) = 2P_M(t|\boldsymbol{x}).$$

$$(9)$$

By the symmetric of permutation we have

$$2\mathbb{E}_{\theta\sim\Theta}[P_W(t|\boldsymbol{x}, \theta)] = \frac{1}{N!}\sum_{V^p\in\Sigma} P_W(t|\boldsymbol{x}, V^p)$$

$$= \frac{1}{N!}\sum_{V^p\in\Sigma}[P_W(t|\boldsymbol{x}, V^p) + P_W(t|\boldsymbol{x}, V^{p^r})]$$

$$= \frac{1}{N!}\sum_{V^p\in\Sigma} 2P_M(t|\boldsymbol{x})$$

$$= 2P_M(t|\boldsymbol{x}).$$

$$(10)$$

Therefore, $\mathbb{E}_{\theta\sim\Theta}[P_W(t|\boldsymbol{x}, \theta)] = P_M(t|\boldsymbol{x})$, which concludes the proof. $\square$

## E   DETAILED EXPERIMENT SETUP

We assess the performance of DiPmark across three critical applications of seq2seq models: text summarization, machine translation, and poetry generation. The experiments are implemented using

the Huggingface library (Wolf et al., 2019), a widely adopted platform for model development and sharing within the NLP community. All experiments are conducted on three Nvidia A6000 GPUs with 48GB of memory. Detecting 1,000 watermarked sentences generated from LLaMA-2 requires only 90 seconds.

**Machine Translation.** For the machine translation task, we utilize the WMT'14 English (En) to Romanian (Ro) dataset, comprising 1,999 examples in the test set. We employ the Multilingual Bart (MBart) model (Liu et al., 2020) along with its official tokenizer.

**Text Summarization.** In the text summarization task, we use the test set from the CNN-DM corpus (Hermann et al., 2015), consisting of 11,490 examples. Our model of choice is BART-large, which encompasses 400 million parameters, and LLaMA-2 with 7 billion parameters.

**Poetry Generation.** For poetry generation, we incorporate the test set from the CNN-DM corpus as part of the generation prompt. Our model selection is LLaMA-2, boasting 7 billion parameters.

**Watermark Setup.** Our experiments primarily compare DiPmark with the Soft watermark introduced by (Kirchenbauer et al., 2023). In the case of DiPmark, we consider various values of $\alpha$ from the set $\{0.3, 0.35, 0.4, 0.45, 0.5\}$. For the Soft watermark (Kirchenbauer et al., 2023), we explore green list bias $\delta$ values from $\{0.0, 1.0, 1.5, 2.0\}$ with a fixed green list separator $\gamma = 0.5$. Texture key generation relies on the most recent five tokens as texture key. For instance, when generating $x_4$ in response to $(x_1, x_2, x_3)$ as the current input to the decoder, the texture key includes $(x_1, x_2, x_3)$, considering the availability of only three tokens. The texture key history resets before generating each batch, ensuring our method's $b$-shot-undetectability given a batch size of $b$. To generate the cipher, we employ SHA-256 as the hash function and a 1024-bit random bitstring as the key $k$. The cipher $\theta$ is sampled from $\Theta$ using hash$(k, \boldsymbol{s})$ as the random seed.

**Evaluation metrics for text quality.** In this part, we introduce the evaluation metrics we used for evaluating the text quality (Section. 6.1).

- **ROUGE score.** For the summarization task, we utilize the ROUGE score (Lin, 2004), which measures n-gram overlap to assess the summary's effectiveness in capturing essential content from reference summaries.

- **BLEU score.** For the machine translation task, we rely on the BLEU score (Papineni et al., 2002), emphasizing the lexical similarity between machine-generated translations and human reference translations.

- **BERTScore.** BERTScore Zhang et al. (2019) computes the similarity of two sentences as a sum of cosine similarities between their tokens' embeddings. We use BERTScore-F1, BERTScore-Precision, and BERTScore-Recall for evaluating both text summarization and machine translation tasks.

- **Perplexity.** In information theory, perplexity is a measurement of how well a probability distribution or probability model predicts a sample. It may be used to compare probability models. A low perplexity indicates the probability distribution is good at predicting the sample. We use perplexity for evaluating both text summarization and machine translation tasks.

**Evaluation metrics for detectability of watermarks.** In this part, we introduce the evaluation metrics we used for evaluating the detectability of watermarks (Sections 6.4 and 6.3).

- **Green token ratio.** Denoted by $L_G(\gamma)$ the number of green tokens in a text sequence with green list separator $\gamma$. The green token ratio is given by $L_G(\gamma)/n - (1 - \gamma)$. This ratio quantifies the bias towards green tokens within the text sequence (see Section 5).

- **z-score.** The z-score of a text sequence $\boldsymbol{x}_{1:n}$ is $\Phi(\gamma, \boldsymbol{x}_{1:n}) := (L_G(\gamma) - (1 - \gamma)n)/\sqrt{n}$. A higher z-score will reduce the false positive rate, where a non-watermarked sequence is detected as watermarked (see Section 5).

- **Type I and II errors.** We generally use true positive rate (TPR), false positive rate (FPR), true negative rate (TNR), and false negative rate (FNR) to evaluate the performance of watermarks on a mixture of watermarked and non-watermarked sentence. FPR measures the Type I error of the hypothesis testing, in which the null hypothesis got rejected when it

**Table 6:** Performance of Machine Translation.

|  | BERT-F1 | BERT-Precision | BERT-Recall | BLEU |
|---|---|---|---|---|
| No Watermark | 0.559±0.003 | 0.545±0.004 | 0.574±0.003 | 21.8±0.3 |
| DiPmark($\alpha$=0.3) | 0.561±0.003 | 0.547±0.004 | 0.575±0.003 | 22.0±0.3 |
| DiPmark($\alpha$=0.35) | 0.562±0.003 | 0.548±0.004 | 0.575±0.003 | 22.1±0.3 |
| DiPmark($\alpha$=0.4) | 0.561±0.003 | 0.547±0.004 | 0.576±0.003 | 21.9±0.3 |
| DiPmark($\alpha$=0.45) | 0.562±0.003 | 0.548±0.004 | 0.576±0.003 | 21.9±0.3 |
| DiPmark($\alpha$=0.5) | 0.562±0.003 | 0.548±0.004 | 0.576±0.003 | 21.8±0.3 |
| Soft($\delta$=0.0) | 0.560±0.003 | 0.545±0.004 | 0.574±0.003 | 21.8±0.3 |
| Soft($\delta$=1.0) | 0.557±0.003 | 0.543±0.004 | 0.572±0.003 | 21.2±0.3 |
| Soft($\delta$=1.5) | 0.550±0.003 | 0.534±0.004 | 0.565±0.003 | 20.4±0.3 |
| Soft($\delta$=2.0) | 0.539±0.003 | 0.523±0.004 | 0.555±0.003 | 19.4±0.3 |

**Table 7:** Performance of Text Summarization.

|  | BERT-F1 | BERT-Precision | BERT-Recall | Perplexity | Rouge-1 | Rouge-2 | Rouge-L |
|---|---|---|---|---|---|---|---|
| No Watermark | 0.3273±0.0008 | 0.3181±0.0009 | 0.3366±0.0010 | 5.021±0.018 | 0.3855±0.0009 | 0.1387±0.0008 | 0.2444±0.0008 |
| DiPmark($\alpha$=0.3) | 0.3279±0.0008 | 0.3187±0.0009 | 0.3372±0.0010 | 5.014±0.018 | 0.3861±0.0009 | 0.1390±0.0008 | 0.2450±0.0008 |
| DiPmark($\alpha$=0.35) | 0.3274±0.0008 | 0.3183±0.0009 | 0.3367±0.0010 | 4.998±0.018 | 0.3856±0.0009 | 0.1389±0.0008 | 0.2449±0.0008 |
| DiPmark($\alpha$=0.4) | 0.3277±0.0008 | 0.3187±0.0009 | 0.3370±0.0010 | 5.001±0.018 | 0.3862±0.0009 | 0.1392±0.0008 | 0.2449±0.0007 |
| DiPmark($\alpha$=0.45) | 0.3269±0.0008 | 0.3178±0.0009 | 0.3361±0.0010 | 5.024±0.018 | 0.3852±0.0009 | 0.1391±0.0008 | 0.2447±0.0008 |
| DiPmark($\alpha$=0.5) | 0.3272±0.0008 | 0.3181±0.0009 | 0.3364±0.0010 | 5.014±0.018 | 0.3859±0.0009 | 0.1396±0.0008 | 0.2450±0.0008 |
| Soft($\delta$=0.0) | 0.3273±0.0008 | 0.3181±0.0009 | 0.3366±0.0010 | 5.021±0.018 | 0.3855±0.0009 | 0.1387±0.0008 | 0.2444±0.0008 |
| Soft($\delta$=1.0) | 0.3237±0.0008 | 0.3137±0.0009 | 0.3338±0.0009 | 5.309±0.019 | 0.3816±0.0009 | 0.1348±0.0008 | 0.2411±0.0007 |
| Soft($\delta$=1.5) | 0.3209±0.0008 | 0.3097±0.0009 | 0.3323±0.0010 | 5.660±0.021 | 0.3793±0.0009 | 0.1317±0.0007 | 0.2379±0.0007 |
| Soft($\delta$=2.0) | 0.3146±0.0008 | 0.3027±0.0009 | 0.3266±0.0009 | 6.241±0.023 | 0.3725±0.0009 | 0.1252±0.0007 | 0.2321±0.0007 |

is actually true. FNR measures the type II error, in which one fails to reject a null hypothesis that is actually false.

# F ADDITIONAL EXPERIMENTS

## F.1 EMPIRICAL VERIFICATION OF DISTRIBUTION-PRESERVING PROPERTY OF DIPMARK

**Settings.** In our evaluation, we assess the performance of DiPmark within the context of two significant applications involving seq2seq models: machine translation (MT) and text summarization (TS). We follow the settings in Hu et al. (2023). For the TS task, our experimentation employs the BART-large model (Liu et al., 2020) in conjunction with the CNN-DM corpus (Hermann et al., 2015) as our designated testing dataset. The MT task, on the other hand, revolves around English-to-Romanian translation. For this purpose, we employ the Multilingual BART (MBart) model (Liu et al., 2020) on the WMT'14 En-Ro corpus. Specifically for DiPmark, we select values for $\alpha$ from the set $\{0.3, 0.35, 0.4, 0.45, 0.5\}$, while for the Soft watermark (Kirchenbauer et al., 2023), we choose green list bias values $\delta$ from the set $\{0.0, 1.0, 1.5, 2.0\}$ alongside a fixed green list separator $\gamma = 0.5$, indicating that 50% of tokens are green while the remainder are red. It is important to note that the Soft watermark with $\delta = 0.0$ is essentially equivalent to no watermark since it does not promote the probability of green list tokens.

A thorough examination of Figure 4, Figure 5, Table 6, and Table 7 reveals a discernible trend. Throughout the range of $\alpha$ values spanning $\{0.3, 0.35, 0.4, 0.45, 0.5\}$, all the metrics associated with machine translation tasks and text summarization tasks maintain a consistent alignment between DiPmark and the original language model. Conversely, an upward adjustment in the $\delta$ values of the Soft watermark distinctly impacts the quality of the text output.

## F.2 DETECTABILITY COMPARISON

**Settings.** We evaluate the detectability of our watermark on text summarization tasks using LLaMA-2. We generate 1,000 examples for each tasks. We also select $\alpha \in \{0.3, 0.35, 0.4, 0.45, 0.5\}$ for

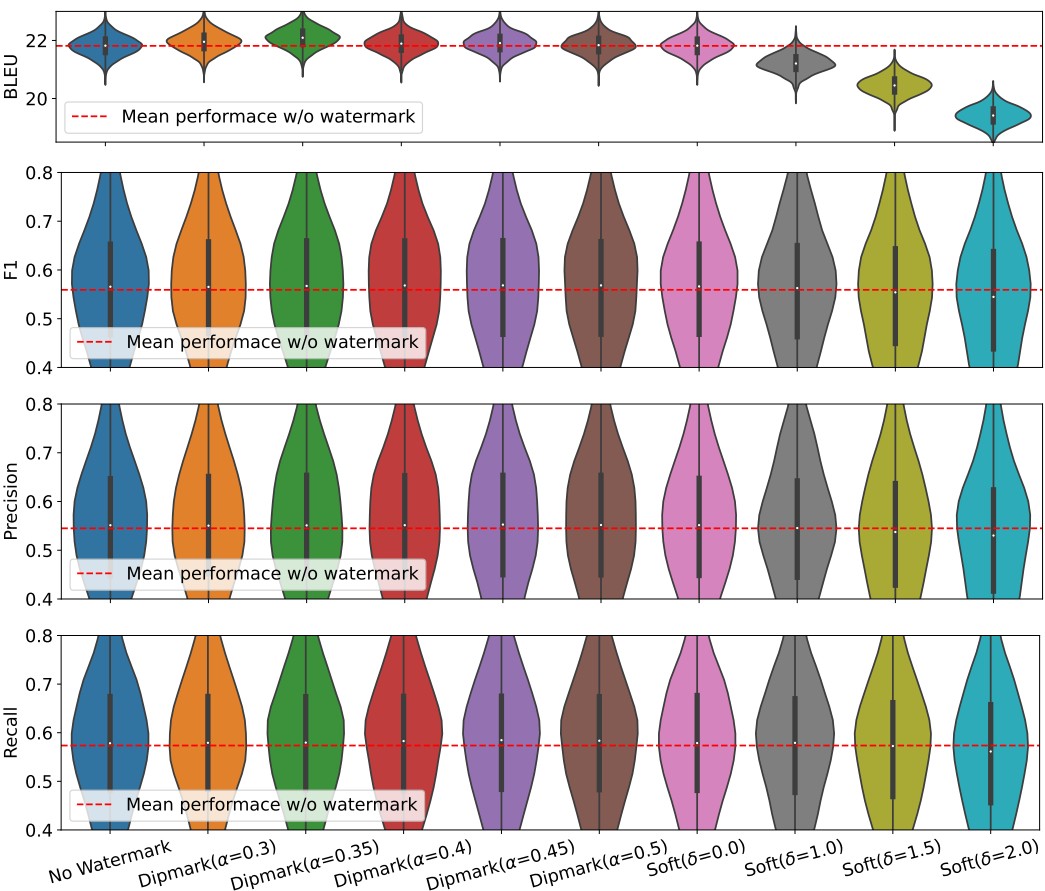

**Figure 4:** Violin plot of Machine Translation performance .

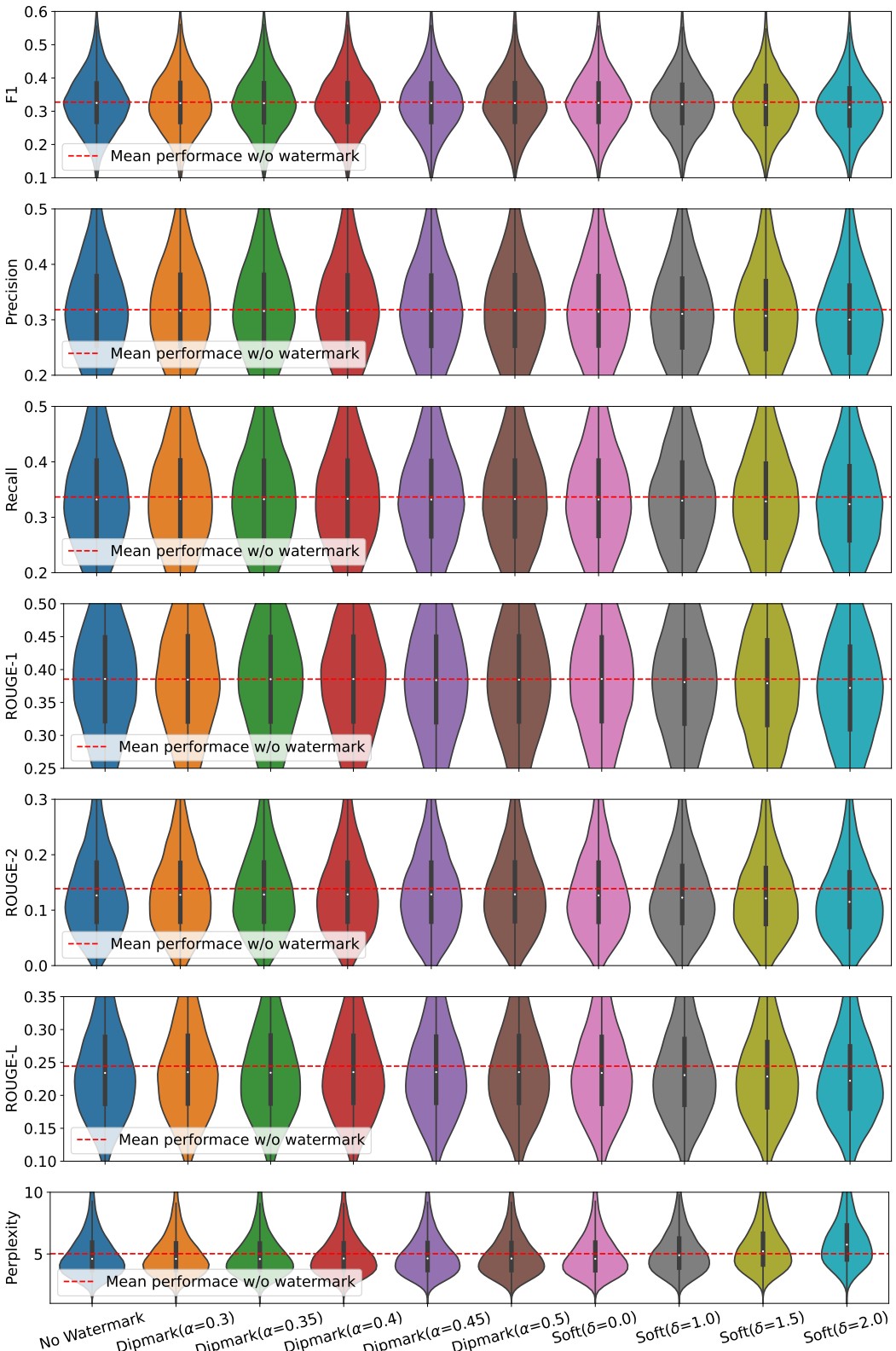

**Figure 5:** Violin plot of Text Summarization performance.

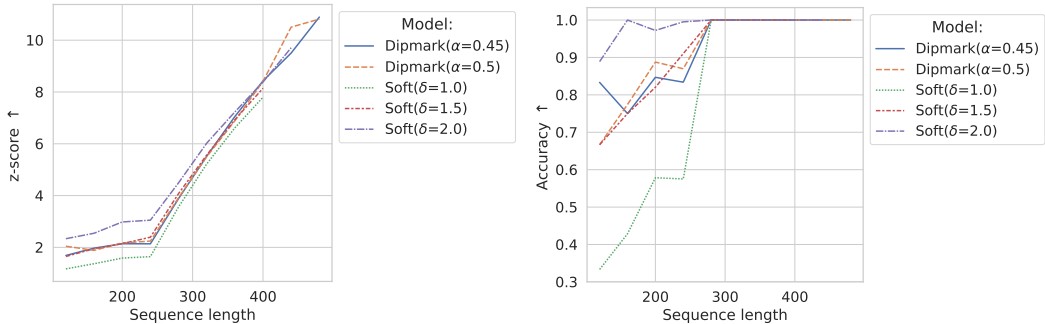

**Figure 6: Left:** Average z-score vs token sequence length with $\gamma = 0.5$ on poetry generation tasks. **Right:** Watermark detection accuracy vs token sequence length with $\gamma = 0.5$ and threshold $z = 1.517$ (false positive rate less than 0.01) on poetry generation tasks.

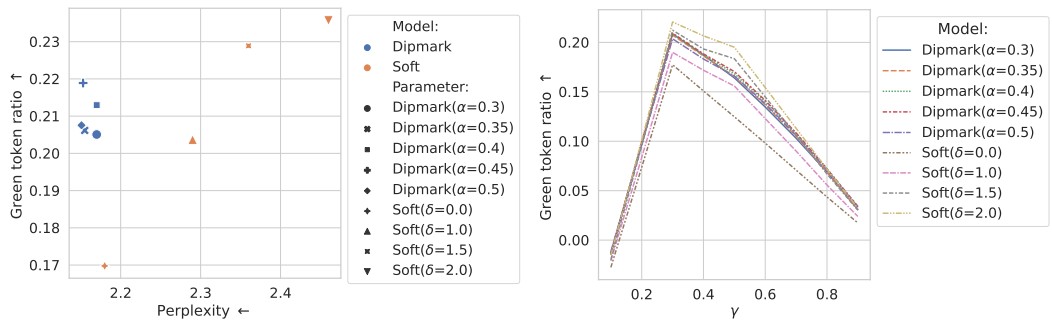

**Figure 7: Left.** Average Perplexity vs Green token rate with $\gamma = 0.5$ on the text summarization task. **Right.** Avg. Green token ratio with different $\gamma$ on the text summarization task.

DiPmark, and $\delta \in \{0.0, 1.0, 1.5, 2.0\}$ and $\gamma = 0.5$ for Soft watermark (Kirchenbauer et al., 2023). During detection, we also use $\gamma = 0.5$. We report the green token ratio (defined in 5), the score of $\Phi(\gamma, \boldsymbol{x})$ (z-score), and the detect accuracy.

**Result analysis.** The results for text summarization are visually depicted in Figure 7 and Figure 8. Broadly speaking, our DiPmark variants with $\alpha = 0.45$ and $0.5$ exhibit performance comparable to that of the Soft watermark with $\delta = 1.5$, where $\delta = 1.5$ corresponds to an augmentation of 1.5 to the green token logits. In Figure 7 (left), it is evident that our DiPmark variants with $\alpha = 0.45$ and $0.5$ yield green token ratios akin to those of the Soft watermark with $\delta = 1.5$ without any discernible degradation in text quality. Figure 7 (right) delves into the impact of different green list separators $\gamma$. Interestingly, for most watermark models, $\gamma = 0.3$ yields the highest green token ratio instead of $\gamma = 0.5$, which may be due to the low entropy characteristic of the text summarization task. In Figure 8 (left) and Figure 8 (right), we present the average z-scores and accuracy metrics relative to sequence length. It is conspicuously observable that longer token sequences tend to facilitate easier detection, in line with our earlier analysis in Section 5.

### F.3 RESILIENCE EVALUATION

We conduct experiments to test the resiliency of the our DiPmark and the Soft watermark in Kirchenbauer et al. (2023). In this context, we use the text summarization tasks with 1,000 generated sequences on LLaMA-2. For resilience evaluation, we manipulating about $\epsilon \in \{0.05, 0.1, 0.2, 0.3\}$ portion of the text tokens through text insertion, text substitution, and text deletion.

**Result Analysis.** Figure 9 elucidates the evolution of the average green token ratio and the average z-score concerning the attack strength parameter $\epsilon$. Notably, both metrics exhibit a diminishing trend as $\epsilon$ increases.

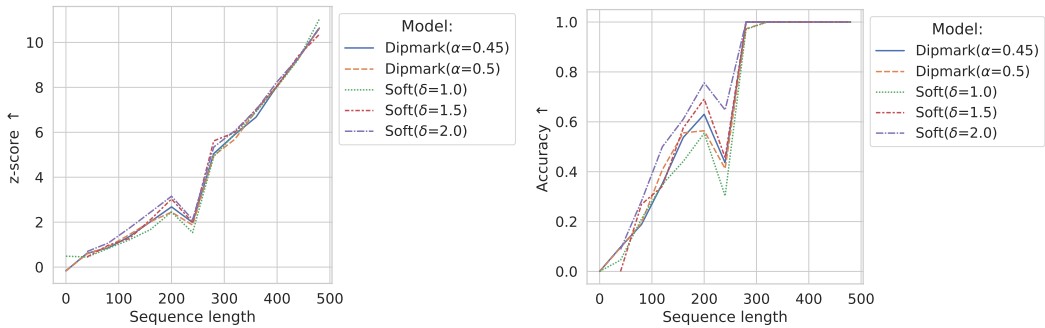

**Figure 8: Left.** Average z-score vs token sequence length with $\gamma = 0.5$ on the text summarization task. **Right.** Avg. best p-score with text length with $\gamma = 0.5$ on the text summarization task.

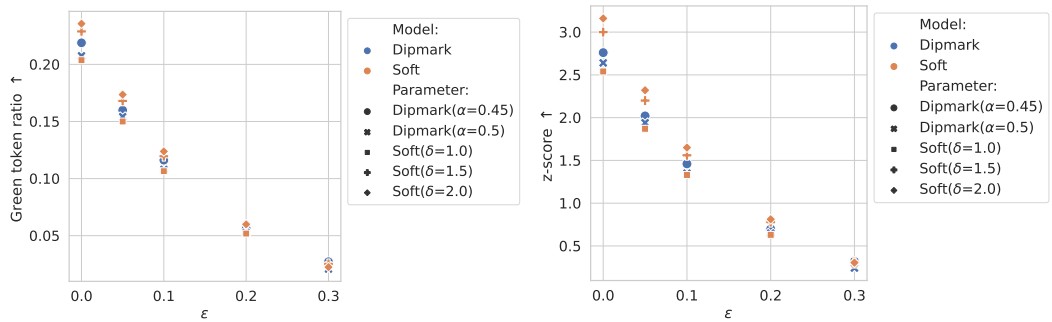

**Figure 9:** Robustness evaluation of DiPmark on text summarization task. **Left.** Average green token ratio w.r.t. portion of perturbation $\epsilon$. **Right.** Average z-score w.r.t. portion of perturbation $\epsilon$.

## G  AN ALTERNATIVE DETECTOR FOR DIPMARK.

Because our objective involves detecting watermarked text, we aim to choose $\gamma$ to maximize the likelihood of null hypothesis rejection. Based on this rationale, we develop a novel detector for watermark detection. Our approach entails a grid search over $\gamma \in [0, 1]$ to identify the optimal z-score, i.e., $\max_{\gamma \in \Gamma} \Phi(\gamma, \boldsymbol{x}_{1:n})$ serves as the definitive detection score for $\boldsymbol{x}$, where $\Gamma = \{\gamma_1, ..., \gamma_l\}$. This process is outlined in Algorithm 3. For the hypothesis test statistic $\max_{\gamma \in \Gamma} \Phi(\gamma, \boldsymbol{x}_{1:n})$, we can establish the following concentration bound:

---

**Algorithm 3** DiPmark detector

1: **Input:** text $\boldsymbol{x}_{1:n}$, watermark key $k$, volume of the token set $N$, permutation generation function $h$, list of separator $\Gamma = \{\gamma_1, ..., \gamma_l\}$, threshold $z$.
2: **for** $\gamma \in \Gamma$ **do**
3:   Initialize the green token indexer of $\gamma$ with 0: $L_G(\gamma) = 0$.
4:   **for** $i = 2, ..., n$ **do**
5:     Generate a texture key $\boldsymbol{s}_i$ based on $\boldsymbol{x}_{1:i-1}$.
6:     Generate the permutation of token set $\theta_i = h(k, \boldsymbol{s}_i)$.
7:     Calculate the list of green tokens via $G = \theta_i[\lceil \gamma N \rceil : N]$.
8:     **if** $x_i \in G$ **then**
9:       Increase the number of green token indexer by 1: $L_G(\gamma) = L_G(\gamma) + 1$.
10:   Calculate the score: $\Phi(\gamma, \boldsymbol{x}_{1:n}) = \frac{L_G(\gamma) - (1-\gamma)n}{\sqrt{n}}$.
11: **return** $\mathbf{1}_{\max_{\gamma \in \Gamma} \Phi(\gamma, \boldsymbol{x}_{1:n}) > z}$.

---

**Theorem G.1.** *Given* $\Gamma = \{\gamma_1, ..., \gamma_l\}$*, under the null hypothesis* $H_0$*, we have*

$$\Pr\left(\max_{\gamma \in \Gamma} \Phi(\gamma, \boldsymbol{x}_{1:n}) \geq t\right) \leq le^{-2t^2}, \quad \forall t \in \mathbb{R}.$$

*Proof.*

$$
\begin{aligned}
\Pr(\max_{\gamma \in \Gamma} \Phi(\gamma, \boldsymbol{x}_{1:n}) < t) &= \prod_{i=1}^{l} \Pr(\max_{\gamma \in \Gamma} \Phi(\gamma, \boldsymbol{x}_{1:n}) < t) \\
&= \prod_{i=1}^{l} (1 - \Pr(\max_{\gamma \in \Gamma} \Phi(\gamma, \boldsymbol{x}_{1:n}) \geq t)) \qquad (11) \\
&\geq (1 - e^{-2t^2})^l \\
&\geq 1 - le^{-2t^2}.
\end{aligned}
$$

Thus, $\Pr(\max_{\gamma \in \Gamma} \Phi(\gamma, \boldsymbol{x}_{1:n}) \geq t) = 1 - \Pr(\max_{\gamma \in \Gamma} \Phi(\gamma, \boldsymbol{x}_{1:n}) < t) \leq le^{-2t^2}.$ $\qquad \square$

Compared to the hypothesis test with a single $\gamma$ and a threshold $z$, our new hypothesis test can achieve the same probability guarantee with a new threshold $z' = \sqrt{z^2 + \log(l)/2}$.

## H  EXAMPLES OF THE WATERMARKED TEXT

We list several examples of the watermarked text generated by LLaMA-2 on the text summarization task. We also report the p-value of the statistal testing using $\Phi(\gamma, \boldsymbol{x}_{1:n})$.

| | Text | p-value |
|---|---|---|
| Prompt | David Cameron likes to project an image as the 'chillaxing' statesman unruffled by political troubles - but this morning his usual demeanour seemed to desert him as he donned a bizarre outfit on the campaign trail. The Prime Minister looked exhausted as he stepped off the sleeper train at Penzance today, ahead of a speech in which he delivered his 'plan for Cornwall'. The Conservatives are battling against their Coalition partners in the South-West as they seek to wrest control of key constituencies from the Liberal Democrats. But Mr Cameron is still trailing Ed Miliband's Labour Party nationally, with just 14 days to go until polling day. Get-up: David Cameron stepped off the sleeper train at Penzance this morning wearing jeans, smart shoes and a navy jacket . Journey: The Prime Minister travelled for eight hours from Paddington station to reach Cornwall . Tired out: The Tory leader did not appear to have enjoyed a good night's sleep on the train and was wearing an unusual combination of solely dark colours . A YouGov poll published this morning put Labour on 34 per cent - one point ahead of the Tories. The Lib Dems trail on 7 per cent with Ukip in third on 14 per cent. Mr Cameron said in an interview this week that he 'always manages to portray a calm smoothness' - however, he looked anything but smooth at Penzance station. The Prime Minister seemed as if he had had trouble sleeping on the eight-hour journey from London's Paddington station. His hair was unkempt and he was wearing an unusual combination of jeans with shiny loafers which might be considered more appropriate for a smart business suit. Destination: He apparently chose the sleeper to avoid taking time out of his schedule to travel . Outfit: Mr Cameron wore a navy jacket, dark blue shirt, jeans and smart black shoes - as well as a red box . Inspiration? Mr Cameron may have been trying to channel the style sense of US President Barack Obama . Mr Cameron sported a navy jacket with a dark blue polo shirt, and set off his outfit with the ultimate status symbol - a red box holding ministerial papers, which only a few dozen people are entitled to carry… | N/A |
| No watermark | 'David Cameron, the Prime Minister of the United Kingdom, was seen wearing an unusual outfit consisting of jeans, smart shoes, and a navy jacket as he stepped off a sleeper train in Penzance, Cornwall. This comes as he embarks on a campaign trail in the South West, where the Conservatives are trying to regain control from the Liberal Democrats. The outfit choice has been likened to that of US President Barack Obama, who is also known for sporting a similar navy and dark blue ensemble. Cameron also discussed his reputation for lacking passion on the campaign trail, attributing it to his hectic schedule and the need to project a calm image. | 0.969 |
| Soft($\delta$=1.0) | "David Cameron, the Prime Minister of the UK, wore an unusual outfit (jeans, loafers, and a navy jacket) while campaigning in Cornwall, leading some to question his 'calm smoothness' persona. He also discussed his love of navy outfits and joked that he thinks William Hague would make a good James Bond. | 4.7e-06 |
| Soft($\delta$=1.5) | 'David Cameron, the Prime Minister of the UK, wore an unusual outfit (jeans, loafers, and a navy jacket) while campaigning in Cornwall, which seemed to reflect his exhausted state. He also claimed to always manage to portray a calm smoothness publicly, although he appeared to be anything but smooth today. Additionally, he expressed support for William Hague as the next James Bond and admitted to enjoying Game of Thrones. | 5.4e-09 |
| Soft($\delta$=2.0) | "David Cameron, the Prime Minister of the UK, wore an unusual outfit (jeans and loafers) while traveling from London to Cornwall, where he gave a speech about his party's plans for the region. He also talked about his love for navy blue clothing and joked about William Hague becoming the next James Bond. | 2.2e-05 |
| DiPmark($\alpha$=0.45) | "David Cameron, the Prime Minister of the UK, wore an unusual outfit (jeans and smart shoes) while traveling on a sleeper train to Cornwall for a campaign event. He also used a selfie stick during a visit to a radio station and took photos with staff. In his speech, he mentioned that he had a rough night's sleep on the train and that he always tries to portray a calm, smooth image.\nMr. Cameron has been known for his love of navy outfits and often wears dark blue when off duty. He has also backed his colleague William Hague to be the next James Bond, citing his fitness and yoga abilities. | 7.6e-30 |
| DiPmark($\alpha$=0.5) | "David Cameron, the Prime Minister of the UK, wore an unusual outfit (jeans and smart shoes) while traveling on a sleeper train to Cornwall for a campaign event. He also used a selfie stick during a visit to a radio station and took photos with staff. In his speech, he mentioned that he had a rough night's sleep on the train and that he always tries to portray a calm, smooth image.\nMr. Cameron has been known for his love of navy outfits and often wears dark blue when off duty. He has also backed his colleague William Hague to be the next James Bond, citing his fitness and yoga abilities. | 7.6e-30 |

**Figure 10:** Examples of the watermarked text generated by LLaMA-2 on text summarization tasks.

| | Text | p-value |
|---|---|---|
| Prompt | Coach Leonardo Jardim believes Monaco's march to the Champions League quarter-finals is proof that clubs do not have to splash the cash in order to achieve European success. Four years have passed since Dmitry Rybolovlev's takeover prompted a slew of big-name arrivals at the Stade Louis II, with the likes of Radamel Falcao, James Rodriguez and Joao Moutinho signed for hefty transfer fees. But the Russian billionaire has tightened the purse strings in recent seasons and, with most of Monaco's expensive signings subsequently moving on, Jardim has been forced to nurture young talent after finding his options in the transfer market somewhat restricted. Leonardo Jardim addresses his players at training  ahead of Champions League match with Juventus . The Monaco coach insists he is proud after his side reached the quarter-finals without big investment . Big money signings James Rodriguez and Radamel Falcao (right) have left the Stade Louis II . Monaco's new-found financial prudence is beginning to yield results, however, with Les Rouges et Blancs third in Ligue 1 and still in with a chance of qualifying for the semi-finals of European football's premier club competition for the first time in 11 years. The principality club host Juventus at the Stade Louis II on Wednesday looking to overturn a 1-0 deficit from the quarter-final first leg and Jardim, who replaced Claudio Ranieri last June, is proud of what he has achieved in his 10 months at the helm. 'We went from the worst qualified team in the fourth pot, to winning the group, to knocking out a European giant in Arsenal (in the last 16),' he told BeIN Sports' The Ligue 1 Show. 'Right now, you could say that Monaco's coach is very proud of his players. We've shown that we don't need to make big investments to go far in this competition. Dimitar Berbatov is convinced Monaco can reach Champions League semi-final at the expense of Juventus… | N/A |
| No watermark | Leonardo Jardim believes Monaco's Champions League success is proof that clubs don't have to spend big to succeed in Europe. Monaco have reached the quarter-finals without making any big signings, instead relying on developing young talent. Jardim is proud of what he has achieved in his 10 months at the helm, and believes the team has shown they can go far in the competition without big investments. Monaco face Juventus in the quarter-finals, and will be without striker Lacina Traore and midfielder Tiemoue Bakayoko due to injury. | 0.68 |
| Soft($\delta$=1.0) | Monaco coach Leonardo Jardim has credited the clubs ability to achieve European success without splashing the cash on big-name signings. Despite the club's financial constraints, Jardim has been able to nurture young talent and lead the team to the Champions League quarter-finals. Monaco host Juventus in the second leg of the quarter-finals on Wednesday, with the principality club looking to overturn a 1-0 deficit. The teams have met once before in the competition, with Monaco winning 2-0 in the 2004 semi-finals. | 9.8e-08 |
| Soft($\delta$=1.5) | Monaco coach Leonardo Jardim has expressed pride in his team's performance in the Champions League quarter-finals after overcoming Arsenal and now facing Juventus, despite the club's financial constraints and lack of big name signings. Jardim believes that his team has shown that they don't need to spend big to achieve success in the competition, and he has been able to nurture young talent at the club. Monaco will be without two key players for the match against Juventus, but Jardim remains optimistic about his team's chances. | 1.5e-11 |
| Soft($\delta$=2.0) | Monaco coach Leonardo Jardim credits the team's success in the Champions League to developing young talent rather than overspending in the transfer market, resulting in the club reaching the quarterfinals without relying heavily on big-name signings. | 6.9e-26 |
| DiPmark($\alpha$=0.45) | Monaco coach Leonardo Jardim has expressed pride in his team's performance in reaching the Champions League quarter-finals without investing heavily in big-name signings. Despite the club's Russian owner Dmitry Rybolovlev tightening the purse strings in recent seasons, Jardim has been able to nurture young talent and achieve success with limited resources. Monaco face Juventus in the quarter-finals and will need to overcome a 1-0 deficit from the first leg to progress to the semi-finals. | 1.0e-39 |
| DiPmark($\alpha$=0.5) | Monaco coach Leonardo Jardim has expressed pride in his team's performance in reaching the Champions League quarter-finals without investing heavily in big-name signings. Despite the club's Russian owner Dmitry Rybolovlev tightening the purse strings in recent seasons, Jardim has been able to nurture young talent and achieve success with limited resources. Monaco face Juventus in the quarter-finals and will need to overcome a 1-0 deficit from the first leg to progress to the semi-finals. | 1.0e-39 |

**Figure 11:** Examples of the watermarked text generated by LLaMA-2 on text summarization tasks.

| | Text | p-value |
|---|---|---|
| Prompt | One in five American women are willing to undergo dangerous beauty treatments in order to achieve the ideal look, despite the risks that these procedures pose to their health. According to a new study, while just over half of women worry about the long term damage of beauty treatments, nearly a fifth would still pursue a treatment to get the right look – even it it proved hazardous to their health. Seven per cent, meanwhile, have actually had allergic reactions. The survey, conducted by beauty research organization\xa0LQS and Associates, looked at the lengths 1,000 American women go to in order to enhance their appearances or copy a celebrity, and the potentially disastrous consequences they might face in doing so, including hair loss, skin swelling, and overly painful procedures. The cost of beauty: Women often don't realize the dangers of salon treatments before sitting in the styling chair . Respondents cited the hairstyles of actresses Jennifer Aniston and Sandra Bullock, as well as signer Jennifer Lopez, as celebrity favorites, with 53 per cent also noting that they get skin, beauty, and hair ideas from pictures and videos of famous personalities. But LQS and Associates warns against attempts to emulate these looks on a regular basis. A-listers like Beyonce, Paris Hilton, and Selena Gomez all sport weaves and extensions, but their beauty regimen is not always adaptable to the average person. 'Many of those glamorous looks can do more harm than good, leading to long term problems,' said LaQue Gushon-Harris, president of LQS and Associates. 'Unfortunately, most women are not even aware of the repercussions.'… | N/A |
| No watermark | According to a new survey, almost a fifth of American women are willing to undergo dangerous beauty treatments despite the risks to their health. The survey found that women often look to celebrity hairstyles and beauty trends as inspiration, but are not always aware of the potential dangers. Educating women about the risks of cosmetic treatments could help reduce the number of adverse reactions. | 0.17 |
| Soft($\delta$=1.0) | According to a survey, 1 in 5 American women are willing to undergo dangerous beauty treatments despite the risks to their health. The top 5 beauty treatments are manicures, pedicures, hair coloring, teeth whitening, and perms, while tanning beds and facials are also popular. Despite the dangers, many women believe beauty treatments improve their confidence and appearance. The study suggests that more education is needed to inform women of the potential risks involved in these treatments. | 6.8e-04 |
| Soft($\delta$=1.5) | According to a survey, 1 in 5 American women are willing to undergo dangerous beauty treatments despite the risks to their health. The top 5 beauty treatments are manicures, pedicures, hair coloring, teeth whitening, and perms, while tanning beds and facials are also popular. Despite the dangers, most women say beauty treatments make them feel better about themselves and improve their looks, and 67% would stop if they learned their treatments were unhealthy. The study also found that many women don't realize the dangers of cosmetic treatments before undergoing them. | 3.1e-02 |
| Soft($\delta$=2.0) | According to a survey, 1 in 5 American women are willing to undergo dangerous beauty treatments despite the risks to their health, with Jennifer Aniston and Sandra Bullock being the top two celebrity hairstyles emulated. Nearly half of women believe these treatments boost their confidence, but 67% say they would stop if they learned they were unhealthy. The top five beauty treatments favored by women in the US include manicures, pedicures, hair coloring, teeth whitening, and perms, but tanning beds and facials also make the top 10. Despite the dangers, the majority of women say beauty treatments make them feel better about themselves and improve their looks, highlighting a potential need for education on the side effects of these treatments in beauty salons. | 5.0e-05 |
| DiPmark($\alpha$=0.45) | According to a survey of 1,000 American women, nearly one in five are willing to undergo dangerous beauty treatments to achieve the ideal look, despite the risks to their health. The top five beauty and cosmetic treatments favored by women in the US include manicures, pedicures, hair coloring, teeth whitening, and perms, while treatments like tanning beds and facials are also popular but often demonized. Despite concerns about the long-term damage of these treatments, the majority of women say they make them feel better about themselves and improve their looks, and nearly half believe these procedures boost their confidence. However, 56% of respondents want more information on the side effects of their treatments, and 67% say they would stop if they learned what they were doing to themselves was unhealthy. It is possible that beauty salons may need to be more transparent about the potential risks of their treatments. | 2.6e-04 |
| DiPmark($\alpha$=0.5) | According to a survey of 1,000 American women, nearly one in five are willing to undergo dangerous beauty treatments to achieve the ideal look, despite the risks to their health. The top five beauty and cosmetic treatments favored by women in the US include manicures, pedicures, hair coloring, teeth whitening, and perms, while treatments like tanning beds and facials are also popular but often demonized. Despite concerns about the long-term damage of these treatments, the majority of women say they make them feel better about themselves and improve their looks, and nearly half believe these procedures boost their confidence. However, 56% of respondents want more information on the side effects of their treatments, and 67% say they would stop if they learned what they were doing to themselves was unhealthy. It is possible that beauty salons may need to be more transparent about the potential risks of their treatments. | 2.5e-04 |

**Figure 12:** Examples of the watermarked text generated by LLaMA-2 on text summarization tasks.

| | Text | p-value |
|---|---|---|
| Prompt | An eight-grade girl accused of conspiring with her soldier boyfriend by text message to have her mother killed must remain in adult jail while awaiting trial, a judge. A defense petition to return 14-year-old Jamie Silvonek to the juvenile facility where she was initially sent after the body of 54-year-old Cheryl Silvonek was discovered last month was denied by Lehigh County Judge Maria Dantos on Friday. District Attorney Jim Martin had opposed the petition on behalf of the teenager, who is charged as an adult with homicide and criminal conspiracy. Her boyfriend, Caleb Barnes, 20, who is from El Paso, Texas, but was stationed at Fort Meade, Maryland, is charged with homicide. JamieSilvonek, the eighth-grader accused of conspiring with her soldier boyfriend by text message to have her mother killed must remain in an adult jail while awaiting trial, it was ruled on Friday . Cheryl Silvonek's body was found with stab wounds in a shallow grave about 50 miles northwest of Philadelphia, and her vehicle partially submerged in a pond a few miles from the family home. Jamie Silvonek was sent to the county jail this month after she was charged as an adult and is in the women's housing unit, away from older inmates, county officials said. Authorities said Barnes and the teenager met in October, when she was 13, but she told him she was 17. Records show the 14-year-old was upset that her mother had shown her boyfriend her passport as proof that Silvonek was not 17. The teen's mother found out about their relationship in early March and ordered them to end it. A day later, the teen told Barnes in a text message, 'I want her gone,' police said. On March 14, Cheryl Silvonek learned that Barnes intended to take her daughter to a concert in Scranton. 'He is not coming to this house,' Cheryl Silvonek texted to her daughter, who had taken the SAT college entrance exam that morning, according to a police affidavit. The eighth-grader sent her mother a text in response begging her to let them go. 'Mom, if I can't go I'm going to have a mental breakdown,' she said, according to the affidavit. 'I need to go and I need to be with my friends… | N/A |
| No watermark | A 14-year-old girl is accused of conspiring with her 20-year-old boyfriend to kill her mother, and the judge has denied her petition to return to a juvenile facility while awaiting trial. The girl's attorney claims she was under the control of her boyfriend and is seeking a psychological evaluation to determine if she should be tried in juvenile court | 0.75 |
| Soft($\delta$=1.0) | A 14-year-old girl is accused of conspiring with her 20-year-old soldier boyfriend to kill her mother after the mother discovered their relationship, which was illegal because the girl was underage.\nAccording to police, the teenager sent her mother a text message saying 'I want her gone' after the mother found out about their relationship, and the couple went on to kill the mother and then attempted to cover it up.\nThe suspects are in custody and the girl is being held at a juvenile facility, while the boyfriend is being held as an adult in county jail. | 0.53 |
| Soft($\delta$=1.5) | A 14-year-old girl is accused of conspiring with her 20-year-old soldier boyfriend to have her mother killed. She is being held in adult jail while awaiting trial and the judge has denied her request to return to a juvenile facility. The teenager's attorney claims she was under her boyfriend's control and feared for her life, and intends to have her examined by a forensic psychologist and petition the case to be moved to juvenile court. | 4.1e-05 |
| Soft($\delta$=2.0) | A 14-year-old girl is accused of conspiring with her 20-year-old soldier boyfriend to have her mother killed. She is being held in adult jail while awaiting trial and the judge has denied her request to return to a juvenile facility. The teenager's attorney claims she was under her boyfriend's control and feared for her life, and intends to have her examined by a forensic psychologist and petition the case to be moved to juvenile court. | 6.4e-17 |
| DiPmark($\alpha$=0.45) | A 14-year-old girl is accused of conspiring with her 20-year-old soldier boyfriend to kill her mother by text message. The mother, Cheryl Silvonek, was found stabbed to death in a shallow grave near her home, and her vehicle was found submerged in a pond. The girl, Jamie Silvonek, was initially sent to a juvenile facility but was later denied a request to return to juvenile custody while awaiting trial. A judge ruled that she must remain in adult jail while awaiting trial. | 6.3e-24 |
| DiPmark($\alpha$=0.5) | A 14-year-old girl is accused of conspiring with her 20-year-old soldier boyfriend to kill her mother by text message. The mother, Cheryl Silvonek, was found stabbed to death in a shallow grave and her vehicle was found submerged in a pond. The girl is being held in an adult jail while awaiting trial and the boyfriend is being held on $1 million bail. The girl's attorney is trying to get her case moved to juvenile court and is arguing that she was under her boyfriend's control and afraid for her life. | 1.6e-18 |

**Figure 13:** Examples of the watermarked text generated by LLaMA-2 on text summarization tasks.

| | Text | p-value |
|---|---|---|
| Prompt | A hotel owner believes he has found the perfect solution for revitalising his flagging business - by converting it into a swingers' paradise. Jon Huxley, 46, hopes to cash in on the Fifty Shades of Grey effect and attract guests from the gay and swinging communities at his hotel Westward Ho! in Folkestone, Kent. He plans to install sex swings, bondage rooms and dungeons and have rooms of differing sizes to cater for couples and multiple groups. Jon Huxley, standing outside his hotel Westward Ho! in Folkestone, Kent, is looking at 'developing' his premises to cater for swingers . He hopes the craze in Fifty Shades of Grey, starring Jamie Dornan as Christian Grey and Dakota Johnson as his object of affection Anastasia Steele, can help transform his business. He said: 'We have plans to attract tourists from the gay and swinging community for short breaks and weekends of adult fun. 'Obviously a lot of equipment will need to be installed like swings, bondage rooms, red room of pain and dungeons of delight. 'We would be having these events each weekend and accepting tourist guests during the week. 'Food and drink will be provided as part of an all-inclusive hotel package and it would be a civilised and friendly environment.' Mr Huxley will be hoping to recreate scenes like this from the hit film Fifty Shades of Grey . Clothing will be optional and privacy for our guests will therefore need to be ensured. 'It is not intended to be a seedy or dirty business. It is a respected clientele who enjoy taking their clothes off.' Mr Huxley said he has decided to appeal to the swinging community after rowing with the council over the… | N/A |
| No watermark | A hotel owner in Folkestone, Kent, UK, is looking to transform his hotel into a swingers\' paradise, in an attempt to revitalize his business. He plans to install sex swings, bondage rooms, and dungeons, and cater to guests from the gay and swinging communities. Locals are not happy about the proposal, with some expressing concerns about noise and anti-social behavior. The hotel owner believes that the "Fifty Shades of Grey effect" will help drive business to his hotel | 0.99 |
| Soft($\delta$=1.0) | A hotel owner in Folkestone, Kent, UK, is looking to transform his hotel into a swingers' paradise, in an attempt to revitalize his business. He plans to install sex swings, bondage rooms, and dungeons, and cater to guests from the gay and swinging communities. Locals are unclear how to feel about the proposal, and the hotel owner is aware that he will need to obtain the necessary licenses and consent from the council. | 7.6e-10 |
| Soft($\delta$=1.5) | A hotel owner in the UK is planning to turn his hotel into a swingers\' paradise, complete with bondage rooms, dungeons, and sex swings. He hopes to cash in on the "Fifty Shades of Grey" effect and attract guests from the gay and swinging communities. The hotel is a Grade-II listed building, and the owner will need to obtain licenses and approvals from the local council before making any changes. The plans have generated controversy and mixed reactions from local residents, with some expressing concern about the potential impact on the community. | 0.67 |
| Soft($\delta$=2.0) | A hotel owner in the UK is planning to renovate his hotel to cater specifically to swingers and the gay and swinging communities, complete with bondage rooms, dungeons, and sex swings. This follows the success of the Fifty Shades of Grey movie franchise, which has fuelled interest in bondage and BDSM.\nThe move is intended to breathe new life into the hotel's flagging business, but it remains to be seen how the local community will react to this unconventional plan. The hotel owner is aware of the potential controversy and has sought to assuage concerns by pointing out that guests will be required to make advance bookings and that privacy will be ensured. However, it is unclear how the hotel's neighbors and local authorities will react to this unique venture. | 1.9e-03 |
| DiPmark($\alpha$=0.45) | A hotel owner is looking to convert his hotel into a swingers\' paradise, complete with bondage rooms, dungeons, and "red room of pain". He hopes to cash in on the popularity of Fifty Shades of Grey and attract guests from the gay and swinging communities. However, the local council has given him a hard time over the proposal and he may need to secure additional licenses and permits. | 1.1e-13 |
| DiPmark($\alpha$=0.5) | A hotel owner is looking to convert his hotel into a swingers\' paradise, complete with bondage rooms, dungeons, and "red room of pain". He hopes to cash in on the popularity of Fifty Shades of Grey and attract guests from the gay and swinging communities. However, the local council has given him permission to use the building as a hotel but has not granted him a premises licence to sell alcohol, and he will need to obtain listed building consent for any internal works.\nFirst of all, I love the fact that the owner is so optimistic about his business idea, but at the same time, I also feel a bit worried for him. This is just a weird and risky idea, and I\'m not sure how well it will go over with the locals or with other guests at the hotel. I mean, I can see how some people might be interested in this kind of thing, but I also think that a lot of people might be put off by it. And then there\'s the issue of the legality of it all, which seems pretty complicated. I think the owner might have bitten off more than he can chew with this one. | 1.0e-03 |

**Figure 14:** Examples of the watermarked text generated by LLaMA-2 on text summarization tasks.

