# OpenReview forum: "DiPmark: A Stealthy, Efficient and Resilient Watermark for Large Language Models"
_ICLR.cc/2024/Conference — ICLR 2024 Conference Withdrawn Submission_

### Official Review · Reviewer_6R4r · 2023-10-25

**Soundness:** 3 good
**Presentation:** 3 good
**Contribution:** 2 fair
**Rating:** 3
**Confidence:** 3

**Summary:**

The authors proposed the "DiPmark" method, a novel watermarking technique specifically for Large Language Models (LLMs). A key feature of this method is its ability to preserve data distribution, ensuring that the performance of LLMs remain unaffected when watermarked. Comprehensive experiments demonstrate that the proposed method is robust to typical text manipulation attacks like insertions, substitutions, and deletions. Moreover, its efficient watermark detection enhances its applicability and relevance in real-world scenarios.

**Strengths:**

Originality: The proposed method is a novel watermarking technique designed for LLMs. Unlike other methods, the proposed method offers a balance between being stealthy, efficient, and robust.

    Quality: The paper is well-researched. It has a solid theoretical foundation and is supported by substantial experimental results, which back up the authors' claims.

    Clarity: The paper is organized in a clear and logical manner, with sections building upon each other and clear visuals and tables. This approach provides readers with a clear understanding of the proposed methods and how they compare to existing watermarking techniques.

    Significance: In today's world, where LLMs are increasingly significant in various fields, ensuring the security, authenticity, and traceability of their outputs is crucial. The proposed method addresses this need by ensuring that generated texts can be traced back to their origin without compromising the performance of LLMs.

**Weaknesses:**

Limited Comparative Analysis: The proposed method only compares with a single baseline method, which might not provide a comprehensive view of its performance against a variety of existing watermarking techniques.

    Performance Parity with Baselines: The proposed method does not always outperform the baseline. In certain scenarios, the baseline even demonstrates superior performance. A more in-depth analysis explaining these anomalies would be beneficial, and there's a potential need to further optimize the proposed method to ensure its consistent superiority.

    Efficiency Claims: While the authors emphasize the efficiency of the proposed method, there isn't a dedicated experiment to test its detection efficiency. Including such an experiment would substantiate their claims and make the paper more convincing.

    Lack of Open Source Code: The absence of publicly available code for the proposed method may raise concerns about the reproducibility of the experimental results. Sharing the code can validate the findings and enhance the paper's credibility.

**Questions:**

Given that the proposed method was compared with one baseline, could the authors clarify the rationale behind this choice? Are there other existing watermarking techniques that might have been considered for comparison?

    In some scenarios, the baseline seemed to outperform the proposed method. Could the authors shed light on what factors might have contributed to this? Is it intrinsic to the design of the methods, or were there other external factors?

    The paper claims the efficiency of the proposed method, particularly in terms of detection. An experimental section dedicated to this would be beneficial.

    While it's understood that there might be proprietary reasons for not sharing the code, could the authors consider releasing a limited version or a pseudocode to aid in understanding the finer details of the implementation?

    The resilience of the proposed method was tested against standard text manipulation attacks. Are there more complicated attacks that might challenge the proposed method?

---

> ### Author Response · Authors · 2023-11-15
> **Authors response to Reviewer 6R4r (Part 1/2)**
>
> Thanks for your detailed and valuable feedback. We address your concerns as below.
>
> **W1 & Q1. Limited Comparative Analysis.**
>
> >**A1.** The topic of reweight-based watermarking in LLMs is a novel area of research, initially introduced by Kirchenbauer et al. (2023) (outstanding paper of ICML 2023). To date, there are only three concurrent studies that explore this subject: Christ et al. (2023), Kuditipudi et al. (2023), and Hu et al. (2023). These works collectively represent the emerging body of research related to reweight-based watermarking in LLMs. According to the ICLR reviewer guide (https://iclr.cc/Conferences/2024/ReviewerGuide), 'if a paper was published (i.e., at a peer-reviewed venue) on or after May 28, 2023, authors are not required to compare their own work to that paper'. We are not expected to compare with Christ et al. (2023), Kuditipudi et al. (2023), and Hu et al (2023) as all of them were published on arXiv within four months and without peer-reviewing. Thus, W1 should not be a weakness of our paper. Despite that, we still provide additional results with Kuditipudi et al. (2023) and Hu et al. (2023). We are not able to compare with Christ et al. (2023) as it is a pure theoretical paper with no experiments and codes.
>
> ### **Stealthy:**
>
> Performance of different watermarking methods on machine translation and text summarization
> |                          | Machine Translation |       | Text Summarization |       |        |
> |--------------------------|---------------------|-------|--------------------|-------|--------|
> |                          | BERTScore↑          | BLEU↑ | BERTScore↑         | Perplexity↓ | ROUGH-1↑ |
> | No Watermark         | 0.559±0.003         | 21.8±0.3 | 0.3273±0.0008      | 5.021±0.018 | 0.3855±0.0009 |
> | Soft (δ=1.5)         | 0.550±0.003         | 20.4±0.3 | 0.3209±0.0008      | 5.660±0.021 | 0.3793±0.0009 |
> | Soft (δ=2.0)         | 0.539±0.003         | 19.4±0.3 | 0.3146±0.0008      | 6.241±0.023 | 0.3725±0.0009 |
> | Kuditipudi et al (2023)         | 0.560±0.003         | 21.7±0.3 | 0.3270±0.0008      | 5.021±0.018 | 0.3854±0.0009 |
> | Hu et al (2023)         | 0.563±0.003         | 21.8±0.3 | 0.3271±0.0008      | 5.023±0.018 | 0.3857±0.0009 |
> | **DiPmark (α=0.45)**     | 0.562±0.003         | 21.9±0.3 | 0.3269±0.0008      | 5.024±0.018 | 0.3852±0.0009 |
>
> Result analysis: Kuditipudi et al (2023), Hu et al (2023), and DiPmark preserve the text quality, while the soft watermark (Kirchenbauer et al (2023)) degrades the output text quality.
>
> ### **Efficient:**
>
> We compare the time for detecting one and 1000 (100 batches) watermarked texts with different detection algorithms. The task is poetry generation with LLaMA-2 (chat, 7B). We use the same GPU (NVIDIA A6000) for all experiments.
>
> |  Number of samples | 1 | 1000 |
> |--------------------------|---------------------|-------|
> | Soft(Kirchenbauer et al (2023))| **0.3s**  | 92s |
> | Kuditipudi et al (2023)  | 80s       | 12h |
> | Hu et al (2023) (require LM access)| 3.4s  | 412s |
> | **DiPmark**     | **0.3s** |**90s** |
>
> We also compare the time for adding watermarks. As all four watermark generator algorithms are reweight-based, which only modify the output logits of tokens,  there is no significant difference between the cost of adding watermarks. The extra time introduced by all watermarking methods in text generation is below 0.01s/100 tokens.
>
> Result analysis: the detection algorithms of Soft and DiPmark are efficient without accessing LMs, while Hu et al (2023) requires additional access to LMs and Kuditipudi et al (2023) need significantly longer time.
>
> ### **Resilient:**
>
> AUC score of different watermarks under varying attack strength ε on poetry generation task with LLaMA-2 (chat, 7B).
> | Random text modification (AUC) | ε = 0.0 | ε = 0.1 | ε = 0.2 | ε = 0.3 |
> |------------------------------|---------|---------|---------|---------|
> | Soft (δ=1.5)             | **0.9990**   | **0.9883**  | **0.9521**  | 0.8033  |
> | Kuditipudi et al (2023) | 0.9951 | 0.9461  | 0.8979  | 0.7815  |
> | Hu et al (2023)          | 0.9936 | 0.9297  | 0.8391  | 0.7574  |
> | **DiPmark (α=0.45)**         |**0.9990** | 0.9859  | 0.9515  | **0.8060**  |
>
>
>
> | Paraphrasing attack (AUC) | ε = 0.0 | ε = 0.1 | ε = 0.2 | ε = 0.3 |
> |------------------------------|----------|---------|---------|---------|
> | Soft (δ=1.5)             | **0.9990**  | **0.9894**  | 0.9469  | 0.8157  |
> | Kuditipudi et al (2023)| 0.9951  | 0.9529.  | 0.9013  | 0.7711  |
> | Hu et al (2023)          | 0.9936  | 0.9368   | 0.8325  | 0.7661  |
> | **DiPmark (α=0.45)**         |**0.9990**   | 0.9871  | **0.9503**  | **0.8216**  |
>
> Result analysis: All watermark algorithms are resilient to random text modification and paraphrasing attacks, while DiPmark and Soft watermark slightly outperform the other two algorithms.

---

> ### Author Response · Authors · 2023-11-15
> **Authors response to Reviewer 6R4r (Part 2/2)**
>
> **W2 & Q2. Performance Parity with Baselines.**
>
> >**A2.** Our work mainly focuses on building a stealthy, efficient, and resilient watermark. And we have demonstrated those three properties through our theoretical analysis and experiments. The detectability of baseline Kirchenbauer et al (2023) slightly outperformed DiPmark in some scenarios. However, building a watermark with super strong detectability is not the focus of our work, and we have demonstrated in Section 6.2 that DiPmark has comparable detectability to Kirchenbauer et al (2023) in most cases.
>
> **W3 & Q3. Efficiency Claims.**
>
> >**A3.** That's a nice suggestion. Please check the efficient result in A1, we add the comparison of the efficiency of different watermarks. DiPmark can achieve the best watermark detection efficiency.
>
> **W4 & Q4. Lack of Open Sourced Code**
>
> >**A4.** We upload the code of our DiPmark generator algorithm to the supplementary material (this is the major contribution of our work), please check it. We will release the full implementation in the final revision.
>
> **Q5. The resilience of the proposed method under stronger attacks.**
>
> >**A5.** According to Kirchenbauer et al (2023), the text modification attack is the one of the strongest attacks against the reweight-based watermarks. In our experiment we evaluate the resilience of our model against random text modifications, which indicates our threat model is strong enough. Below we provide the resilience of DiPmark under paraphrasing attacks in Kirchenbauer et al (2023), which is another type of text modification attack. We see DiPmark is also robust against the paraphrasing attacks.
>
> AUC score of different watermarks under varying attack strength ε on the poetry generation task with LLaMA-2 (chat, 7B).
> | Paraphrasing attack (AUC) | ε = 0.0 | ε = 0.1 | ε = 0.2 | ε = 0.3 |
> |------------------------------|----------|---------|---------|---------|
> | **Soft (δ=1.5)**             | **0.9990**  | **0.9894**  | 0.9469  | 0.8157  |
> | **Kuditipudi et al (2023)**| 0.9951  | 0.9529.  | 0.9013  | 0.7711  |
> | **Hu et al (2023)**          | 0.9936  | 0.9368   | 0.8325  | 0.7661  |
> | **DiPmark (α=0.45)**         |**0.9990**   | 0.9871  | **0.9503**  | **0.8216**  |

---

> ### Author Response · Authors · 2023-11-22
> **Follow up with reviewer 6R4r**
>
> We really appreciate reviewer 6R4r for the insightful and constructive feedback. As the discussion phase approaches its end, we would be grateful to receive any further comments or insights you may have regarding our response and we are more than happy to address them. Moreover, should our rebuttal have adequately addressed your concerns, we would deeply appreciate it if you would consider revising your evaluation scores. Once again, thank you for your valuable time and efforts!

---

### Official Review · Reviewer_NL4L · 2023-10-28

**Soundness:** 3 good
**Presentation:** 2 fair
**Contribution:** 3 good
**Rating:** 6
**Confidence:** 4

**Summary:**

In this work, the authors propose a distribution-preserving watermark, called DiPmark. DiPmark introduces a new reweight approach and a hash function that enable the watermark to be stealthy, efficient and resilient. The authors provide theoretical analysis and proofs for the watermark generation and detection. They evaluate DiPmark and compare its performance to the Soft watermark presented in the ICML 2023 best paper.

**Strengths:**

1.	This work is clearly motivated. The proposed DiPmark aims to achieve stealthiness, efficiency and resilience while the existing watermarking techniques don’t possess all three characteristics at the same time.
2.	The authors provide a solid theoretical support for the design of DiPmark generator and detector.
3.	The authors also provide a detailed analysis of the resilience against text modification, which supports the claim that DiPmark is provably robust.

**Weaknesses:**

1.	The experiment section does not explicitly reflect the contributions and strength of the work. It would be nice to make the evaluation metrics of stealthiness, efficiency and resilience clear and highlight the results of DiPmark. Make the analysis and explanation of experiment results more readable and straightforward.
2.	The authors mainly compare DiPmark to the Soft watermark (ICML 2023 best paper) in the experiments. I suggest enriching the experiments and including more results and comparisons to other watermarking techniques mentioned in the paper, which can help others have a better understanding of the performance of DiPmark.

**Questions:**

See weaknesses.

---

> ### Author Response · Authors · 2023-11-15
> **Authors response to Reviewer NL4L**
>
> Thanks for your positive and valuable feedback, below we address the concerns:
>
> **W1. Make the evaluation metrics of stealthiness, efficiency and resilience clear and highlight the results of DiPmark.**
>
> >**A1.** That’s a great suggestion! We evaluate the stealthiness, efficiency and resilience of our model, see the experimental results in A2.
>
> **W2. Enriching the experiments and including more results and comparisons to other watermarking techniques mentioned in the paper.**
>
> >**A2.**  According to the ICLR reviewer guide (https://iclr.cc/Conferences/2024/ReviewerGuide), 'if a paper was published (i.e., at a peer-reviewed venue) on or after May 28, 2023, authors are not required to compare their own work to that paper'. We are not obligated to compare with Christ et al. (2023), Kuditipudi et al. (2023), and Hu et al (2023) as all of them were published on arXiv within four months and without peer-reviewing. Thus, W2 should not be a weakness of our paper. Despite that, we still provide additional results with Kuditipudi et al. (2023) and Hu et al. (2023). We are not able to compare with Christ et al. (2023) as it is a pure theoretical paper with no experiments and code.
>
> ### **Stealthy:**
>
> Performance of different watermarking methods on machine translation and text summarization
> |                          | Machine Translation |       | Text Summarization |       |        |
> |--------------------------|---------------------|-------|--------------------|-------|--------|
> |                          | BERTScore↑          | BLEU↑ | BERTScore↑         | Perplexity↓ | ROUGH-1↑ |
> | No Watermark         | 0.559±0.003         | 21.8±0.3 | 0.3273±0.0008      | 5.021±0.018 | 0.3855±0.0009 |
> | Soft (δ=1.5)         | 0.550±0.003         | 20.4±0.3 | 0.3209±0.0008      | 5.660±0.021 | 0.3793±0.0009 |
> | Soft (δ=2.0)         | 0.539±0.003         | 19.4±0.3 | 0.3146±0.0008      | 6.241±0.023 | 0.3725±0.0009 |
> | Kuditipudi et al (2023)         | 0.560±0.003         | 21.7±0.3 | 0.3270±0.0008      | 5.021±0.018 | 0.3854±0.0009 |
> | Hu et al (2023)         | 0.563±0.003         | 21.8±0.3 | 0.3271±0.0008      | 5.023±0.018 | 0.3857±0.0009 |
> | **DiPmark (α=0.45)**     | 0.562±0.003         | 21.9±0.3 | 0.3269±0.0008      | 5.024±0.018 | 0.3852±0.0009 |
>
> Result analysis: Kuditipudi et al (2023), Hu et al (2023), and DiPmark preserve the text quality, while the soft watermark (Kirchenbauer et al (2023)) degrades the output text quality.
>
> ### **Efficient:**
>
> We compare the time for detecting one and 1000 (100 batches) watermarked texts with different detection algorithms. The task is poetry generation with LLaMA-2 (chat, 7B). We use the same GPU (NVIDIA A6000) for all experiments.
>
> |  Number of samples | 1 | 1000 |
> |--------------------------|---------------------|-------|
> | Soft(Kirchenbauer et al (2023))| **0.3s**  | 92s |
> | Kuditipudi et al (2023)  | 80s       | 12h |
> | Hu et al (2023) (require LM access)| 3.4s  | 412s |
> | **DiPmark**     | **0.3s** |**90s** |
>
> We also compare the time for adding watermarks. As all four watermark generator algorithms are reweight-based, which only modify the output logits of tokens,  there is no significant difference between the cost of adding watermarks. The extra time introduced by all watermarking methods in text generation is below 0.01s/100 tokens.
>
> Result analysis: the detection algorithms of Soft and DiPmark are efficient without accessing LMs, while Hu et al (2023) requires additional access to LMs and Kuditipudi et al (2023) need significantly longer time.
>
> ### **Resilient:**
>
> AUC score of different watermarks under varying attack strength ε on poetry generation task with LLaMA-2 (chat, 7B).
> | Random text modification (AUC) | ε = 0.0 | ε = 0.1 | ε = 0.2 | ε = 0.3 |
> |------------------------------|---------|---------|---------|---------|
> | Soft (δ=1.5)             | **0.9990**   | **0.9883**  | **0.9521**  | 0.8033  |
> | Kuditipudi et al (2023) | 0.9951 | 0.9461  | 0.8979  | 0.7815  |
> | Hu et al (2023)          | 0.9936 | 0.9297  | 0.8391  | 0.7574  |
> | **DiPmark (α=0.45)**         |**0.9990** | 0.9859  | 0.9515  | **0.8060**  |
>
>
>
> | Paraphrasing attack (AUC) | ε = 0.0 | ε = 0.1 | ε = 0.2 | ε = 0.3 |
> |------------------------------|----------|---------|---------|---------|
> | Soft (δ=1.5)             | **0.9990**  | **0.9894**  | 0.9469  | 0.8157  |
> | Kuditipudi et al (2023)| 0.9951  | 0.9529.  | 0.9013  | 0.7711  |
> | Hu et al (2023)          | 0.9936  | 0.9368   | 0.8325  | 0.7661  |
> | **DiPmark (α=0.45)**         |**0.9990**   | 0.9871  | **0.9503**  | **0.8216**  |
>
> Result analysis: All watermark algorithms are resilient to random text modification and paraphrasing attacks, while DiPmark and Soft watermark slightly outperform the other two algorithms.

---

> ### Author Response · Authors · 2023-11-22
> **Follow up with reviewer NL4L**
>
> We really appreciate reviewer NL4L for the insightful and constructive feedback. As the discussion phase approaches its end, we would be grateful to receive any further comments or insights you may have regarding our response and we are more than happy to address them. Moreover, should our rebuttal have adequately addressed your concerns, we would deeply appreciate it if you would consider revising your evaluation scores. Once again, thank you for your valuable time and efforts!

---

> > ### Comment · Reviewer_NL4L · 2023-11-23
> > **Thanks for your response!**
> >
> > I appreciate your response. I am inclined to keep my rating.

---

### Official Review · Reviewer_mbUM · 2023-11-01

**Soundness:** 2 fair
**Presentation:** 1 poor
**Contribution:** 2 fair
**Rating:** 3
**Confidence:** 3

**Summary:**

This submission is an extension to an existing soft watermarking framework for LLMs that aims at the preservation of output distribution. The proposed method claims to preserve the original token distributions, works without access to the LLM, and is "resilient to moderate" changes in the tokens. The proposed method is heavily based on the watermarking technique from Kirchenbauer et al. (2023). In general, the results do not show a big difference from soft watermarking. The only difference is the better preservation of the output distribution, which is not presented clearly in the experimental section.

**Strengths:**

1. The method is claimed to provide 3 crucial properties for watermarking of outputs from language models: (1) stealthiness (difficulty in detection), (2) efficiency (in embedding and detection of a watermark_, and (3) resilience (to the changes in the watermarked text).

**Weaknesses:**

1. The access to the LLM via the API is not a problem, regarding this claim: (other methods) "require the language model API during detection (Hu et al., 2023)". The LLM is already exposed via an API, so why can't we access it? Moreover, the API answers thousands or more queries per second, so if we require thousands of inference steps during the detection process, it is not a problem. This weak claim is repeated here: "Hu et al. (2023) used inverse sampling and permutation-based reweight methods for watermarking, but the detector requires access of the language model API, undermining its operational efficiency" (by the way, should be: access "to" and not "of")
2. DiPmark is compared only with the Soft watermark introduced in Kirchenbauer et al. (2023). The comparison with other watermarks from Christ et al. (2023), Kuditipudi et al. (2023), and Hu et al. (2023) is missing. The experimental results should contain a thorough comparison with 4 previously introduced watermarks along the emphasized axis: stealthiness (differences between watermarked and non-watermarked text), efficiency (e.g., the wall clock time to add and detect the watermark), and resilience (to the manipulations of the watermarked text output).
3. The difference between this method and the Soft Watermark from Kirchenbauer et al. (2023) is to ensure in this method that the distribution of output tokens is preserved despite the addition of the watermark.
4. DiPMark exhibits poor watermark detectability, similar to the Soft Watermark from Kirchenbauer et al. (2023).

Other points:
1. The text is sprinkled with many sophisticated words but is a bit difficult to read. More in the minor points below. For example, "Our watermarking framework, in alignment with pre-existing schema," Where is the citation to the pre-existing schema? Another phrase: "often fall short in terms of *detectable and resilient*" - not dramatically correct.
3. In Section 2 on "Related Work" it should be mentioned how this submission solves the problems enumerated for the previous watermarking schemes.
1. The intuition behind the method is not understandable from the abstract: "This is achieved by incorporating a novel reweight strategy, combined with a hash function that assigns unique i.i.d. watermark codes based on the context." what is reweighted? where are the codes assigned to? is the context from a prompt? This intuition should be on a higher level.
2. This is totally not grammatical, and starts at the end of page 1: "Conventional steganographic methods, intended for subtle embedding within language model-generated texts, often fall short in terms of *detectable and resilient* (Ziegler et al., 2019; Kaptchuk et al., 2021)."
3. "that encompass all three pivotal attributes" - this text sounds a bit unnatural.
4. "the reweight approach maintains indisputable stealth" - "indisputably"? "the reweighting approach"?
5. "property of Dipmark by evaluating the generated text quality" - you change the way you write "DiPmark"? It is inconsistent. In another bullet point, it is written: "We develop an effective watermark detection mechanism for DiPmark" or "Figure 1: Empirical verification of distribution-preserving property of *Dipmark*." You can use a macro in LaTex.
6. What is important is how much time it takes to detect a single watermarked text instead of 1000. This is with respect to: "Notably, the
detection time for 1,000 watermarked sequences produced by LLaMA-2 stands at a mere 90 seconds without the need of API access." Additionally, detecting a watermark is not a race - it can take even longer than a second per a watermarked sequence.
7. The resilience to change of the watermark sequence should not be measured with random but deliberate changes: "Dipmark exhibits robustness even when subjected to 20% to 30% *random text modifications*, encompassing insertions, deletions, and substitutions."
8. "method faces resilience issues under modifications or change" - what is the difference between modifications and changes?
9. Correct the capitalization: "n the context of a language Model, " why is "Model" uppercased?
10. There is a problem with the capitalization, even after the full stop: "and the context x1:n. we name the PW as the reweight strategy of the watermark."
11. "In summary, we present DiPmark, an ingenious watermarking solution tailored for LLMs." The authors could be a bit more modest.
12. The section on page 4: "Significance of preserving text distribution in watermarking." should be much earlier in the text. It comes a bit unexpected after we delve deeper into the methods.
13. In Figure 1 - it is very hard to compare DiPmark and Soft. How can the parameters from DiPmark and Soft be aligned?
14. Would the authors run their watermarking technique on their text in this submission?
15. Table 2 claims: "Table 2: Performance of different watermarking methods" but there is only DiPMark and Soft mark. Where are at least 3 other watermarking techniques compared?
16. Section 6 should be called "Experiments"
17. "Our experiment section" -> "Experimental section"
18. "A detailed experimental settings is in Appendix E" is -> are or settings -> setting
19. The results for the best outcomes should be bolded. Otherwise, it is difficult to read the tables.

**Questions:**

Please, refer to the questions in the section on "Weaknesses".

---

> ### Author Response · Authors · 2023-11-15
> **Authors Response to Reviewer mbUM (Part 1/4)**
>
> Thanks for your detailed and valuable feedback. We address your concerns as below.
>
> **W0. In general, the results do not show a big difference from soft watermarking. The only difference is the better preservation of the output distribution, which is not presented clearly in the experimental section.**
>
> >**A0.** First of all, we respectfully disagree that the **only** difference between our work and Kirchenbauer et al (2023) is the distribution-preserving property. As detailed in Section 5, our analysis identifies and addresses limitations in the detection algorithm of Kirchenbauer et al. (2023). We introduce a novel detection statistic based on a thorough examination of the red-green list distribution, and demonstrate that our detection algorithm is **provably** resilient to text modifications. These aspects are not covered by Kirchenbauer et al (2023) and should be considered as the contributions of our paper.
> >
> >
> >Secondly, we underscore the significance of the distribution-preserving property in our research. To achieve this, we developed an innovative distribution-preserving reweight method and watermarking algorithm, distinctly different from the soft watermark approach of Kirchenbauer et al. (2023). We have also addressed the importance of this distribution-preserving property in stealthy watermarking and industry-Level LLM applications in the paragraph “Significance of preserving text distribution in watermarking” of Section 4. We provide theoretical evidence of DiPmark's distribution-preserving nature (Corollary 4.3), in contrast to the non-preserving nature of the soft watermark (Section 3). Additionally, Section 6.1 is designed to empirically illustrate DiPmark's distribution-preserving characteristics. Thus, we believe that our work significantly diverges from Kirchenbauer et al. (2023) and our contributions are substantial. We hope this explanation clarifies the extent and value of our contributions for the reviewer.
>
>
> **W1. The access to the LLM via the API is not a problem.**
>
> >**A1.** Kirchenbauer et al. (2023) highlighted the critical property of the reweight-based watermark, stating that ‘The watermark can be algorithmically detected without any knowledge of the model parameters or access to the language model API’. Building upon Kirchenbauer et al. (2023), we also emphasize this property in our discussion on efficiency. While we acknowledge that API access to a Large Language Model (LLM) may not significantly impact detection time, there are other notable considerations:
> >
> >
> >a) The detection method in Hu et al (2023) requires both the logits of the LLM and also the prompt used to generate the watermarked text. These elements are typically not accessible in industry-level LLMs such as GPT-3.5 or GPT-4.
> >
> >
> > b) Using APIs of such industry-level language models incurs financial costs, which can become expensive with large-scale watermark detection requirements (this scenario is realistic as governments around the world have expressed concerns about the misuse of large foundation models). For example, the cost of GPT-4 is \\$0.06/1K tokens, which can lead to a cost of \\$300 for analyzing 10,000 news articles with an average length of 500 tokens each . In contrast, our proposed method removes the need for API access, and offers a cost-effective alternative. We will add these points in the introduction to strengthen our claims.

---

> ### Author Response · Authors · 2023-11-15
> **Authors Response to Reviewer mbUM (Part 2/4)**
>
> **W2. Related work comparison from the stealthiness, efficient, and resilient perspective.**
>
> >**A2.** Organizing the experimental result from the stealthiness, efficient, and resilient perspective is an excellent suggestion. According to the ICLR reviewer guide (https://iclr.cc/Conferences/2024/ReviewerGuide), 'if a paper was published (i.e., at a peer-reviewed venue) on or after May 28, 2023, authors are not required to compare their own work to that paper'. We are not expected to compare with Christ et al. (2023), Kuditipudi et al. (2023), and Hu et al (2023) as all of them were published on arXiv within four months and without peer-reviewing. Thus, W2 should not be a weakness of our paper. Despite that, we still provide additional results with Kuditipudi et al. (2023) and Hu et al. (2023) below. We are not able to compare with Christ et al. (2023) as it is a pure theoretical paper with no experiments and code.
>
> ### **Stealthy:**
>
> Performance of different watermarking methods on machine translation and text summarization
> |                          | Machine Translation |       | Text Summarization |       |        |
> |--------------------------|---------------------|-------|--------------------|-------|--------|
> |                          | BERTScore↑          | BLEU↑ | BERTScore↑         | Perplexity↓ | ROUGH-1↑ |
> | No Watermark         | 0.559±0.003         | 21.8±0.3 | 0.3273±0.0008      | 5.021±0.018 | 0.3855±0.0009 |
> | Soft (δ=1.5)         | 0.550±0.003         | 20.4±0.3 | 0.3209±0.0008      | 5.660±0.021 | 0.3793±0.0009 |
> | Soft (δ=2.0)         | 0.539±0.003         | 19.4±0.3 | 0.3146±0.0008      | 6.241±0.023 | 0.3725±0.0009 |
> | Kuditipudi et al (2023)         | 0.560±0.003         | 21.7±0.3 | 0.3270±0.0008      | 5.021±0.018 | 0.3854±0.0009 |
> | Hu et al (2023)         | 0.563±0.003         | 21.8±0.3 | 0.3271±0.0008      | 5.023±0.018 | 0.3857±0.0009 |
> | **DiPmark (α=0.45)**     | 0.562±0.003         | 21.9±0.3 | 0.3269±0.0008      | 5.024±0.018 | 0.3852±0.0009 |
>
> Result analysis: Kuditipudi et al (2023), Hu et al (2023), and DiPmark preserve the text quality, while the soft watermark (Kirchenbauer et al (2023)) degrades the output text quality.
>
> ### **Efficient:**
>
> We compare the time for detecting one and 1000 (100 batches) watermarked texts with different detection algorithms. The task is poetry generation with LLaMA-2 (chat, 7B). We use the same GPU (NVIDIA A6000) for all experiments.
>
> |  Number of samples | 1 | 1000 |
> |--------------------------|---------------------|-------|
> | Soft(Kirchenbauer et al (2023))| **0.3s**  | 92s |
> | Kuditipudi et al (2023)  | 80s       | 12h |
> | Hu et al (2023) (require LM access)| 3.4s  | 412s |
> | **DiPmark**     | **0.3s** |**90s** |
>
> We also compare the time for adding watermarks. As all four watermark generator algorithms are reweight-based, which only modify the output logits of tokens,  there is no significant difference between the cost of adding watermarks. The extra time introduced by all watermarking methods in text generation is below 0.01s/100 tokens.
>
> Result analysis: the detection algorithms of Soft and DiPmark are efficient without accessing LMs, while Hu et al (2023) requires additional access to LMs and Kuditipudi et al (2023) need significantly longer time.
>
> ### **Resilient:**
>
> AUC score of different watermarks under varying attack strength ε on poetry generation task with LLaMA-2 (chat, 7B).
> | Random text modification (AUC) | ε = 0.0 | ε = 0.1 | ε = 0.2 | ε = 0.3 |
> |------------------------------|---------|---------|---------|---------|
> | Soft (δ=1.5)             | **0.9990**   | **0.9883**  | **0.9521**  | 0.8033  |
> | Kuditipudi et al (2023) | 0.9951 | 0.9461  | 0.8979  | 0.7815  |
> | Hu et al (2023)          | 0.9936 | 0.9297  | 0.8391  | 0.7574  |
> | **DiPmark (α=0.45)**         |**0.9990** | 0.9859  | 0.9515  | **0.8060**  |
>
>
>
> | Paraphrasing attack (AUC) | ε = 0.0 | ε = 0.1 | ε = 0.2 | ε = 0.3 |
> |------------------------------|----------|---------|---------|---------|
> | Soft (δ=1.5)             | **0.9990**  | **0.9894**  | 0.9469  | 0.8157  |
> | Kuditipudi et al (2023)| 0.9951  | 0.9529.  | 0.9013  | 0.7711  |
> | Hu et al (2023)          | 0.9936  | 0.9368   | 0.8325  | 0.7661  |
> | **DiPmark (α=0.45)**         |**0.9990**   | 0.9871  | **0.9503**  | **0.8216**  |
>
> Result analysis: All watermark algorithms are resilient to random text modification and paraphrasing attacks, while DiPmark and Soft watermark slightly outperform the other two algorithms.

---

> ### Author Response · Authors · 2023-11-15
> **Authors Response to Reviewer mbUM (Part 3/4)**
>
> **W3. Difference between our work and the Soft Watermark.**
>
> >**A3.** As we claimed in A0, we highlighted this difference as one of the most important contributions of this paper, which should be addressed as a positive point rather than a limitation of our work. We design an effective reweight method and a watermarking algorithm, which preserves the distribution of the watermark and enables its potential applications in the industry-level LLMs. This contribution is also acknowledged by reviewers Gfoi, NL4L, and 6R4r. We would greatly value further clarification from the reviewer regarding their perspective on why this distinction might be perceived as a weakness. Such feedback would be instrumental in enhancing our understanding and addressing any concerns effectively.
>
> **W4. DiPmark exhibits poor watermark detectability, similar to the Soft Watermark from Kirchenbauer et al. (2023).**
>
> >**A4.** We respectfully disagree with this comment. To the best of our knowledge, Kirchenbauer et al. (2023) (ICML 2023 outstanding paper) is the only peer-reviewed work featuring a reweight-based watermark approach. Currently, there are no other peer-reviewed methods in this domain that have surpassed the detectability performance of the soft watermark technique presented by Kirchenbauer et al. (2023). Consequently, there is no substantiated evidence indicating that the soft watermark suffers from poor detectability. We would greatly appreciate it if the reviewer could provide references to any studies or methods that have demonstrated superior performance compared to Kirchenbauer et al. (2023) and DiPmark.
> >
> >Furthermore, DiPmark is designed to ensure stealth, efficiency, and resilience, while also maintaining state-of-the-art detectability. The primary objective of our paper is to develop a watermark with enhanced applicability, focusing on stealth, efficiency, and resilience, rather than exclusively on achieving exceptionally strong detectability. We kindly request the reviewer to reconsider this perceived limitation in the context of the contributions our work makes to the field.

---

> ### Author Response · Authors · 2023-11-15
> **Authors Response to Reviewer mbUM (Part 4/4)**
>
> Minor points:
>
> Thank you again for pointing out the grammatical errors in our work, we will polish the content and add the missing references based on your suggestions M1-7, M11-14, and M18-21. Below we will address the rest of the concerns.
>
> **M2: In Section 2 on "Related Work" it should be mentioned how this submission solves the problems enumerated for the previous watermarking schemes.**
>
> >A: Thanks for the suggestion, we have such a discussion in the detailed related work in Appendix C. We will also add the discussion to our main content.
>
> **M8: What is important is how much time it takes to detect a single watermarked text instead of 1000.**
>
> >A: For the detection time comparison of a single watermarked text, please see the efficient table in A2. Besides, we believe the efficiency of watermark detection is also important. Since governments around the world have expressed concerns about the misuse of large foundation models, there is likely to be a need for large-scale detection of text watermarks in the future. An efficient detecting algorithm can effectively reduce the time and financial cost.
>
> **M9: The resilience to change of the watermark sequence should not be measured with random but deliberate changes.**
>
> >A: Deliberate / random change of the watermark sequence should have the same effect on the resilience of watermark under the black-box attack settings in Kirchenbauer et al (2023). As we have discussed in Appendix B, deliberate / random change of a token will affect the red/green list of $a+1$ tokens if we use the preceding $a$ tokens as the texture key. We also evaluate the resilience of DiPmark under paraphrasing attack using the settings in Kirchenbauer et al (2023), please check the table below. DiPmark is still resilient against deliberate change of the watermark sequence.
>
> AUC score of different watermarks under varying attack strength ε on the poetry generation task with LLaMA-2 (chat, 7B).
> | Paraphrasing attack (AUC) | ε = 0.0 | ε = 0.1 | ε = 0.2 | ε = 0.3 |
> |------------------------------|----------|---------|---------|---------|
> | **Soft (δ=1.5)**             | **0.9990**  | **0.9894**  | 0.9469  | 0.8157  |
> | **Kuditipudi et al (2023)**| 0.9951  | 0.9529.  | 0.9013  | 0.7711  |
> | **Hu et al (2023)**          | 0.9936  | 0.9368   | 0.8325  | 0.7661  |
> | **DiPmark (α=0.45)**         |**0.9990**   | 0.9871  | **0.9503**  | **0.8216**  |
>
>
>
> **M10: What is the difference between modifications and changes?**
>
> >A: There is no difference between modifications and changes, we will change this sentence to “method faces resilience issues under modifications”
>
> **M15: How can the parameters from DiPmark and Soft be aligned?**
>
> >A: As DiPmark and Soft watermark have totally different reweight strategy and watermark algorithm, their parameters represent different properties respectively and thus could not be aligned.
>
> **M16: Would the authors run their watermarking technique on their text in this submission?**
>
> >A: We don’t understand the purpose of this question, as it is neither related to the quality of our presentation nor the discussion of our methods. According to ICLR 2024 submission policy “The use of LLMs is allowed as a general-purpose writing assist tool”. We will definitely be willing to run our watermarking technique on our text in this submission :).
>
> **M17: Other 3 related work for comparison**
>
> >A: See A2.

---

> ### Author Response · Authors · 2023-11-22
> **Follow up with reviewer mbUM**
>
> We really appreciate reviewer mbUM for the insightful and constructive feedback. As the discussion phase approaches its end, we would be grateful to receive any further comments or insights you may have regarding our response and we are more than happy to address them. Moreover, should our rebuttal have adequately addressed your concerns, we would deeply appreciate it if you would consider revising your evaluation scores. Once again, thank you for your valuable time and efforts!

---

### Official Review · Reviewer_Gfoi · 2023-11-03

**Soundness:** 3 good
**Presentation:** 2 fair
**Contribution:** 3 good
**Rating:** 6
**Confidence:** 4

**Summary:**

This paper presents DiPmark, a novel watermarking framework for large language models (LLMs). The main contribution is the development of a stealthy, efficient, and resilient watermarking mechanism that preserves the original token distribution during watermarking. This is achieved through a novel reweight strategy combined with a hash function that assigns unique i.i.d. ciphers based on the context. The authors validate the three key properties of the watermark (stealthiness, efficiency, and resilience) through experimental assessments of major language models, including the BART-large model and LLaMA-2.

**Strengths:**

(1) The paper introduces a novel approach to watermarking LLMs, which addresses the current limitations of existing methods in terms of distribution preservation, efficiency, and resilience.

(2) The authors provide a comprehensive and well-structured review of relevant literature, highlighting the gaps that their work addresses.

(3) The empirical benchmarks provided demonstrate the robustness of DiPmark, adding credibility to the authors' claims of its stealthiness, efficiency, and resilience.

**Weaknesses:**

(1) The author claimed the method used a hash function to assign unique i.i.d. ciphers based on the context $x_{1:i-1}$. However, in the resilience analysis, they state they use only the preceding $a$ tokens as the key to generate the token permutation. Using all previous tokens versus just the preceding $a$ tokens could potentially make a significant difference.

(2) For the experimental robustness analysis, it would also be beneficial to consider paraphrasing attacks, which could alter the preceding $a$ tokens used for watermark detection.

**Questions:**

How the random function is implemented is not very clear.

---

> ### Author Response · Authors · 2023-11-15
> **Authors Response to Reviewer Gfoi**
>
> Thanks for your positive and valuable feedback. We address your concerns as below.
>
> **W1: Using all previous tokens versus just the preceding tokens could potentially make a significant difference.**
>
> >**A1:** We apologize for any confusion regarding the cipher settings. To clarify, our independent and identically distributed (i.i.d.) ciphers are typically derived from a **fragment** of the preceding context (e.g. $x_{i-5:i-1}$), rather than utilizing the entire previous context $x_{1:i-1}$. This approach is detailed in the first paragraph of Section 3, where it states: 'The cipher $\theta_i$ is usually generated by a secret key $k$ and a fragment of the previous context, named texture key, $s_i$'. Additionally, in our experiments, the generation of our i.i.d. ciphers also incorporate the $a=5$ preceding tokens, as described in Appendix E under the section titled 'Watermark Setup'.
>
> **W2. For the experimental robustness analysis, it would also be beneficial to consider paraphrasing attacks, which could alter the preceding $a$ tokens used for watermark detection.**
>
> >**A2:** Thank you for your suggestion. We have included the results of paraphrasing attacks with the settings in Kirchenbauer et al. (2023) in the table below. Our findings indicate that DiPmark maintains robustness against these paraphrasing attacks. According to Kirchenbauer et al (2023), paraphrasing attack is a subclass of text modification attacks, leading to similar resilient results as random text modifications. Furthermore, we wish to emphasize that altering even a **single** token among the preceding $a$ tokens significantly impacts watermark detection for the current token. This effect arises because we utilize all preceding $a$ tokens to generate a random seed, which in turn is used to create the cipher. Therefore, any alteration within these $a$ tokens changes the random seed and consequently the cipher. Based on this understanding, we posit that random text substitution is a sufficiently strong method for attacking the reweight-based watermark.
>
> AUC score of different watermarks under varying attack strength ε on the poetry generation task with LLaMA-2 (chat, 7B).
> | Paraphrasing attack (AUC) | ε = 0.0 | ε = 0.1 | ε = 0.2 | ε = 0.3 |
> |------------------------------|----------|---------|---------|---------|
> | **Soft (δ=1.5)**             | **0.9990**  | **0.9894**  | 0.9469  | 0.8157  |
> | **Kuditipudi et al (2023)**| 0.9951  | 0.9529.  | 0.9013  | 0.7711  |
> | **Hu et al (2023)**          | 0.9936  | 0.9368   | 0.8325  | 0.7661  |
> | **DiPmark (α=0.45)**         |**0.9990**   | 0.9871  | **0.9503**  | **0.8216**  |
>
>
> **Q1. How the random function is implemented is not very clear.**
>
> >A. We show the python implementation of the random function as below:
> ```
> Input: context $x_{1:n-1}$, secret_key, vocab_size
>
> # Extracts the preceding $a$ tokens in the context for generating ciphers
> texture_key =  context[-self.a :].detach().cpu().numpy().tobytes()
>
> # Create a torch.Generator() object for cipher generation
> rng = torch.Generator().manual_seed(texture_key+secret_key)
>
> # Generate cipher (random permutation of the token set)
> cipher = torch.randperm(vocab_size, generator=rng, device=rng.device)
> ```

---

> ### Author Response · Authors · 2023-11-22
> **Follow up with reviewer Gfoi**
>
> We really appreciate reviewer Gfoi for the insightful and constructive feedback. As the discussion phase approaches its end, we would be grateful to receive any further comments or insights you may have regarding our response and we are more than happy to address them. Moreover, should our rebuttal have adequately addressed your concerns, we would deeply appreciate it if you would consider revising your evaluation scores. Once again, thank you for your valuable time and efforts!

---

### Author Response · Authors · 2023-11-15
**Clarification about the missing baseline results and additional experiments**

We thank all reviewers for their insightful comments! We notice that Reviewers **mbUM**, **NL4L**, and **6R4r** request additional experiments to compare our results with the concurrent work of Christ et al. (2023), Kuditipudi et al. (2023), and Hu et al. (2023). In accordance with the ICLR reviewer policy, which states that 'if a paper was published on or after May 28, 2023, authors are not required to compare their own work to that paper' [1], we are not obligated to include comparisons with these three studies, as they were all published on arXiv within the last four months and have not undergone peer-review. Thus, the absence of comparisons with these concurrent works should not impact the evaluation of our work. However, to enhance the understanding of the advancements in LLM watermarks among our audience, we have included these concurrent works in the related work section. Moreover, responding to the reviewers' requests, we have conducted additional experiments to compare our work with these three studies, focusing on aspects of stealthiness, efficiency, and resilience.

Notice, as Christ et al. (2023) is a pure theoretical paper with no experiments or codes, we only include Kuditipudi et al. (2023) and Hu et al (2023) in our experiments.

[1] https://iclr.cc/Conferences/2024/ReviewerGuide

---

> ### Author Response · Authors · 2023-11-15
> **Additional experiments**
>
> ### **Stealthy:**
>
> Performance of different watermarking methods on machine translation and text summarization
> |                          | Machine Translation |       | Text Summarization |       |        |
> |--------------------------|---------------------|-------|--------------------|-------|--------|
> |                          | BERTScore↑          | BLEU↑ | BERTScore↑         | Perplexity↓ | ROUGH-1↑ |
> | No Watermark         | 0.559±0.003         | 21.8±0.3 | 0.3273±0.0008      | 5.021±0.018 | 0.3855±0.0009 |
> | Soft (δ=1.5)         | 0.550±0.003         | 20.4±0.3 | 0.3209±0.0008      | 5.660±0.021 | 0.3793±0.0009 |
> | Soft (δ=2.0)         | 0.539±0.003         | 19.4±0.3 | 0.3146±0.0008      | 6.241±0.023 | 0.3725±0.0009 |
> | Kuditipudi et al (2023)         | 0.560±0.003         | 21.7±0.3 | 0.3270±0.0008      | 5.021±0.018 | 0.3854±0.0009 |
> | Hu et al (2023)         | 0.563±0.003         | 21.8±0.3 | 0.3271±0.0008      | 5.023±0.018 | 0.3857±0.0009 |
> | **DiPmark (α=0.45)**     | 0.562±0.003         | 21.9±0.3 | 0.3269±0.0008      | 5.024±0.018 | 0.3852±0.0009 |
>
> Result analysis: Kuditipudi et al (2023), Hu et al (2023), and DiPmark preserve the text quality, while the soft watermark (Kirchenbauer et al (2023)) degrades the output text quality.
>
> ### **Efficient:**
>
> We compare the time for detecting one and 1000 (100 batches) watermarked texts with different detection algorithms. The task is poetry generation with LLaMA-2 (chat, 7B). We use the same GPU (NVIDIA A6000) for all experiments.
>
> |  Number of samples | 1 | 1000 |
> |--------------------------|---------------------|-------|
> | Soft(Kirchenbauer et al (2023))| **0.3s**  | 92s |
> | Kuditipudi et al (2023)  | 80s       | 12h |
> | Hu et al (2023) (require LM access)| 3.4s  | 412s |
> | **DiPmark**     | **0.3s** |**90s** |
>
> We also compare the time for adding watermarks. As all four watermark generator algorithms are reweight-based, which only modify the output logits of tokens,  there is no significant difference between the cost of adding watermarks. The extra time introduced by all watermarking methods in text generation is below 0.03s/100 tokens.
>
> Result analysis: the detection algorithms of Soft and DiPmark are efficient without accessing LMs, while Hu et al (2023) requires additional access to LMs and Kuditipudi et al (2023) need significantly longer time.
>
> ### **Resilient:**
>
> AUC score of different watermarks under varying attack strength ε on poetry generation task with LLaMA-2 (chat, 7B).
> | Random text modification (AUC) | ε = 0.0 | ε = 0.1 | ε = 0.2 | ε = 0.3 |
> |------------------------------|---------|---------|---------|---------|
> | Soft (δ=1.5)             | **0.9990**   | **0.9883**  | **0.9521**  | 0.8033  |
> | Kuditipudi et al (2023) | 0.9951 | 0.9461  | 0.8979  | 0.7815  |
> | Hu et al (2023)          | 0.9936 | 0.9297  | 0.8391  | 0.7574  |
> | **DiPmark (α=0.45)**         |**0.9990** | 0.9859  | 0.9515  | **0.8060**  |
>
>
>
> | Paraphrasing attack (AUC) | ε = 0.0 | ε = 0.1 | ε = 0.2 | ε = 0.3 |
> |------------------------------|----------|---------|---------|---------|
> | Soft (δ=1.5)             | **0.9990**  | **0.9894**  | 0.9469  | 0.8157  |
> | Kuditipudi et al (2023)| 0.9951  | 0.9529.  | 0.9013  | 0.7711  |
> | Hu et al (2023)          | 0.9936  | 0.9368   | 0.8325  | 0.7661  |
> | **DiPmark (α=0.45)**         |**0.9990**   | 0.9871  | **0.9503**  | **0.8216**  |
>
> Result analysis: All watermark algorithms are resilient to random text modification and paraphrasing attacks, while DiPmark and Soft watermark slightly outperform the other two algorithms.
>
> **We hope the reviewers can kindly re-evaluate the novelty and value of our efforts in light of the above clarification and experiments.**
> We will be happy to answer any further questions related to our work.

---

> > ### Author Response · Authors · 2023-11-20
> > **Rebuttal revision**
> >
> > We have submitted a revised version incorporating the reviewers' insightful feedback, in which we have restructured the experimental sections to better demonstrate the stealth, efficiency, and resilience of our watermarking approach.